# Dosimetric comparison of Monte Carlo, Acuros XB, and anisotropic analytical algorithm for lung cancer plans on halcyon accelerators

**Kainan Shao**[1], **Chaojun Chai**[2], **Yiwei Yang**[2], **Weijun Chen**[1], **Fenglei Du**[2]*

**1** Cancer Center, Department of Radiation Oncology, Zhejiang Provincial People's Hospital (Affiliated People's Hospital), Hangzhou Medical College, Hangzhou, Zhejiang, China, **2** Department of Radiation Physics, Zhejiang Cancer Hospital, Hangzhou Institute of Medicine (HIM), Chinese Academy of Sciences, Hangzhou, Zhejiang, China

* dufl@zjcc.org.cn

**Data availability statement:** All relevant data are within the paper and its Supporting information files.

## Abstract

This study evaluates the differences in application of the RayStation Monte Carlo algorithm (RMC) compared to Acuros XB (AXB) and the Anisotropic Analytical Algorithm (AAA) in Eclipse for conventional radiotherapy planning of lung cancer using the novel ring-shaped Halcyon accelerator. A total of 63 non-small-cell lung cancer patients were retrospectively included, with a prescription dose of 60 Gy delivered in 30 fractions. Radiotherapy plans were initially designed and optimized in RayStation, then recalculated in Eclipse using AXB and AAA to assess algorithmic differences in dose distributions. Analysis of data from 63 patients, combined with simulations using square fields and cylindrical water phantoms, revealed that RMC and AXB achieved high consistency in target dose coverage and conformity. High-dose indicators, such as $D_{2\%}$ and $D_{0.03cc}$, showed close agreement between RMC and AAA, while AXB tended to slightly underestimate peak doses. For prescription dose coverage metrics like $D_{95\%}$ and $D_{98\%}$, the difference between RMC and AXB was less than 1%, whereas AAA exhibited minor degradation. In organ-at-risk dose evaluations, RMC delivered higher doses compared to AXB and AAA, with AAA doses exceeding those of AXB. These findings confirm the dose consistency of RayStation and Eclipse algorithms for use with the Halcyon accelerator. The RayStation Monte Carlo algorithm (RMC) is a viable alternative to AXB, especially in lung cancer cases with high tissue heterogeneity, as target coverage and conformity discrepancies remain within 0.5%. Additionally, recalculation of RMC-optimized plans in Eclipse using AXB resulted in lower organ-at-risk doses. Therefore, recalculations performed before treatment do not compromise the clinical adequacy of volumetric dose evaluations.

**Funding:** This study was supported by Zhejiang Provincial Basic Public Welfare Research Project (No. LGF22H160070) and Zhejiang Medical and Health Project (2022KY673).

**Competing interests:** The authors have declared that no competing interests exist.

## Introduction

Non-small cell lung cancer (NSCLC), accounting for 85% of all lung cancer cases, is a leading cause of cancer-related mortality globally [1]. Among the available treatment modalities, volumetric modulated arc therapy (VMAT) has emerged as a key option for NSCLC radiotherapy due to its advantages in dose distribution, reduced treatment time, and lower radiation exposure to normal tissues [2,3]. Research has demonstrated that reducing metrics such as $V_{20Gy}$ [4,5], $V_{30Gy}$ [6,7], $V_{5Gy}$ [8], and the mean lung dose (MLD) [9,10] significantly lowers the risk of radiation pneumonitis (RP). Consequently, optimizing radiotherapy plans to minimize normal lung tissue exposure while maintaining adequate target dose coverage is of critical clinical importance. However, achieving this balance is challenging due to variations in tumor size and location, as well as the diversity in patient anatomy.

The Halcyon (Varian Medical System, CA) accelerator, a novel ring-gantry design, introduces several innovations over conventional C-arm linacs, including a tungsten-free gantry, dual-layer stacked multi-leaf collimators (MLC), and a flattening filter-free (FFF) mode. These features improve treatment efficiency and normal tissue sparing [11–13], while the ring geometry facilitates rapid cone-beam CT (CBCT) imaging and reduces collision risks [14,15]. Halcyon's daily image-guided radiotherapy (IGRT) capability further enhances setup accuracy and treatment precision, particularly in lung cancer management [16].

RayStation (Raysearch, Stockholm, Sweden) and Eclipse (Varian, Palo Alto, USA) are two widely used treatment planning systems (TPSs) that provide different dose calculation engines. RayStation supports direct machine parameter optimization (DMPO) and offers GPU-accelerated Monte Carlo (MC) dose calculation [17,18], while Eclipse provides Acuros XB (AXB) [19] and the Anisotropic Analytical Algorithm (AAA) [20]. MC is considered the gold standard for dose calculation due to its accurate modeling of particle transport, particularly in heterogeneous tissues such as the lung [21,22]. AXB, by solving the Boltzmann transport equation, achieves near-MC accuracy in heterogeneous regions, whereas AAA, although computationally efficient, may overestimate dose in low-density tissues [23,24].

Although the Halcyon accelerator has demonstrated strong clinical performance, its closed architecture requires that plans generated in RayStation be recalculated in Eclipse prior to approval and delivery in the ARIA record-and-verify system [25]. This workflow preserves the original monitor units (MUs) and plan parameters during Eclipse recalculation to ensure consistency with the RayStation-approved plan report, rather than renormalizing dose, which could alter the clinically documented treatment plan.

Previous studies have compared MC, AXB, and AAA in heterogeneous phantoms or small patient cohorts [26–30]. These studies consistently found that AXB aligns more closely with MC than AAA, particularly in lung tissue. However, most of these investigations were performed on C-arm linacs, lacked cross-TPS evaluation, and did not reflect the specific workflow constraints of Halcyon accelerators, where plans must undergo RayStation-to-Eclipse recalculation before clinical approval and delivery. Saini et al. (2021) modeled Halcyon in RayStation and validated MC against AXB using phantom-based measurements but did not investigate algorithmic consistency in patient plans or the impact of cross-TPS dose recalculation.

Motivated by this unique workflow requirement, we performed the first comprehensive cross-TPS dosimetric validation of RayStation MC (RMC) and Eclipse AXB/AAA for Halcyon, using 63 conventionally fractionated lung cancer patients with complex target geometries, complemented by phantom-based measurements. This study aims to provide practical evidence for algorithmic consistency in a real-world RayStation-to-Eclipse workflow and to guide clinical implementation in institutions adopting this dual-TPS approach.

## Materials and methods

During the commissioning of the Treatment Planning System (TPS), the accelerator model parameters in RayStation were initially adjusted using the Physics module's automatic beam modeling function to match the PDD (percent depth dose) and profile curves measured in a three-dimensional water phantom. The automatically tuned parameters included photon energy spectrum, output factors (with correction for small and large field sizes), multi-leaf collimator (MLC) transmission, MLC tip width and MLC offset, and extra-focal/secondary source contributions, consistent with procedures described in Saini et al. [25]. This process was carried out in collaboration with the manufacturer and supervised by our senior physicist, with subsequent acceptance testing completed. To ensure reliable dose calculation by the three dose engines (RMC, AXB, and AAA) on the same accelerator, we referred to the AAPM Medical Physics Practice Guideline 5.a [31] for testing and verification, ensuring that these algorithms met the required standards and performed stably.

### Cylindrical water phantom test

In this study, dose discrepancies among the three algorithms were initially assessed using a cylindrical water phantom (21 cm in diameter) modeled after the ArcCheck system (Sun Nuclear Corporation, USA), with its internal material overridden to water in the TPS. The experimental setup was as follows: A simple treatment plan was created in both RayStation and Eclipse, using a 10×10 cm$^2$ field size and 300 MUs (assuming 30 fractions as plan dose), with a single gantry angle of 0°, as shown in (Fig 1). The doses were calculated using the RayStation Monte Carlo, AcurosXB, and AAA algorithms, respectively.

RayStation Monte Carlo dose calculation for the phantom study was performed with a statistical uncertainty of 0.2% per plan and a dose grid size of 2 mm, ensuring sufficient accuracy for high-gradient regions. Gamma analysis was then carried out to compare the reference plan with the dose distributions obtained from the MC, AXB, and AAA algorithms. Within

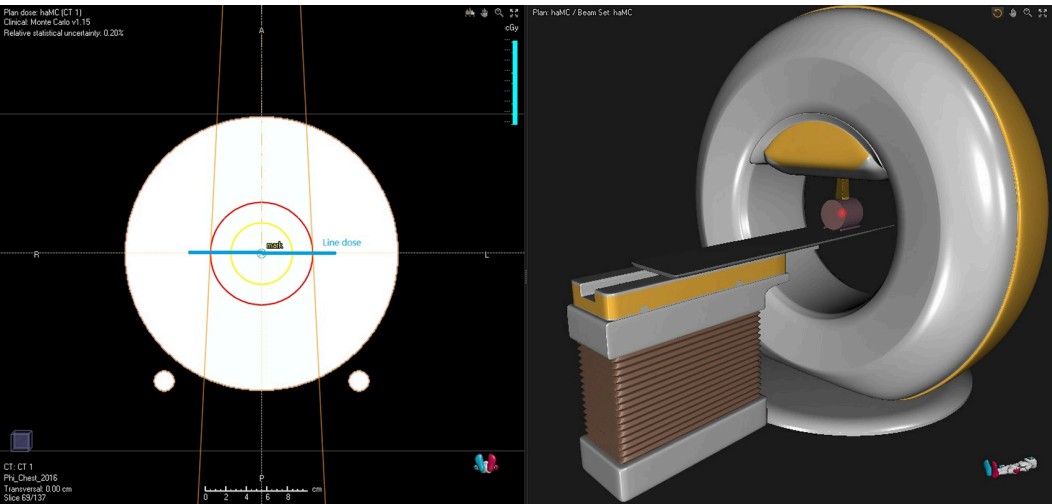

**Fig 1. Illustration of Water Phantom with 10x10 cm² Field Size at 300 MU and Isocenter at the Center of the Cylinder.** Right Panel Shows Room View. The Line dose shown in the image corresponds to the dose profile in Figure 4.

the phantom, two spherical regions of interest (ROIs) with radii of 3 cm and 5 cm were contoured. The 3 cm ROI was placed in the central uniform dose plateau to evaluate dose consistency in the field center, whereas the 5 cm ROI extended toward the periphery of the field to include both the plateau and the surrounding dose gradient. Dose–volume histograms (DVHs) were then extracted from these ROIs to quantify algorithm-specific dose variations in a standardized environment.

This approach was chosen to simulate simplified and standardized experimental conditions under homogeneous media, enabling a direct comparison of the accuracy and consistency of different dose calculation algorithms without the confounding influence of complex anatomical heterogeneities.

## Patient selection

This retrospective study included 63 patients diagnosed with non-small cell lung cancer (NSCLC) who received conventional radiotherapy between 2021 and 2023. All patients were treated with a prescribed dose of 60 Gy in 30 fractions. The inclusion criteria for the study were the following: patients with pathologically confirmed NSCLC who received conventional radiotherapy between 2021 and 2023, with a prescribed dose of 60 Gy/30 fractions. Exclusion criteria included patients who did not receive treatment with the prescribed dose, as well as those without complete treatment plan records. The retrospective study was approved by the Medical Ethics Committee of Zhejiang Provincial People's Hospital (No. QT2024085) and was conducted in accordance with the ethical standards of the Declaration of Helsinki. Patient consent was waived by the Medical Ethics Committee of Zhejiang Provincial People's Hospital due to the anonymity of the data. All patient CT images were anonymized at the time of data collection. The subsequent comparison in this study was conducted using simulated radiotherapy plans rather than actual treatment plans. The study methods and procedures adhered to the Declaration of Helsinki and other relevant regulations. Data for this retrospective analysis were collected and analyzed between August and November 2024.

The cohort was categorized according to tumor stage, with 57. 1% (36/63) of patients in Stage 3 (locally advanced) and 42.8% (27/63) in Stage 4 (distant metastases). Tumor locations varied across the cohort, with 15 cases in the left upper (LU), 27 in the right upper (RU), 5 in the right middle (RM), 7 in the left lower (LL), and 9 in the right lower (RL) lung. Given the clinical characteristics of the tumors, most patients had target volumes that included regional cervical lymph node areas in addition to the primary tumor site. The distribution of clinical target volume (CTV) and planning target volume (PTV) volumes for the cohort is as follows: the mean CTV volume was $169.75 \pm 123.43$ cm³, and the mean PTV volume was $285.4 \pm 161.56$ cm³. To aid in the visualization of these data, a diagram (Fig 2) illustrates the distribution of CTV and PTV volumes in cubic centimeters (cm³). All patients in this study underwent image-guided radiation therapy (IGRT) using the Halcyon accelerator, with daily kV Cone Beam CT (CBCT) for positioning prior to each treatment session, ensuring accurate target localization in accordance with standard operational guidelines for lung cancer radiotherapy. For more comprehensive patient characteristic data, additional details can be found in the Supporting information Sect.

## Simulation setup and target contouring

The CT images of 63 patients were originally obtained during the patients' routine radiotherapy process. The delineation of target volumes and organs at risk (OARs) was based on preexisting clinical data that had been previously approved by experienced radiation oncologists as part of the routine clinical workflow. All patients were scanned using a Brilliance Big Bore

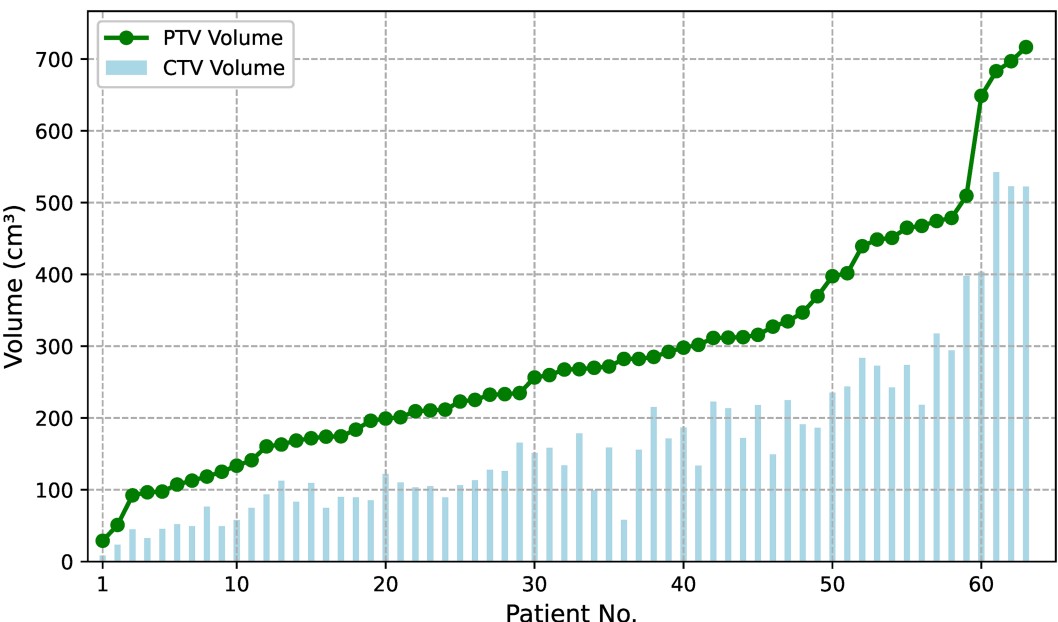

**Fig 2. The distribution of CTV and PTV volumes accross 63 patients.** The vertical axis represents the volume (cm³), and the horizontal axis shows the patient number, sorted by PTV volume.

CT simulator (Philips Healthcare, Best, The Netherlands). The supine position was standardized, with arms extended overhead and immobilization achieved through an integrated fixation board and thermoplastic mask. CT scans covered the region from the upper border of the second cervical vertebra to the lower border of the second lumbar vertebra, with a slice thickness of 5 mm. According to our institution's radiotherapy clinical workflow, for head and neck tumors, the default CT simulation reconstruction slice thickness is 3 mm, and for chest and abdominal scans, due to the larger scanning range, a 5 mm slice thickness is typically used to balance scanning time, image noise, and dose calculation accuracy. If required by the physician, the slice thickness can be adjusted based on specific clinical circumstances. Among the 63 enrolled patients in this study, 55 had their CT scans reconstructed with a 5 mm slice thickness, while the remaining 8 had their CT scans reconstructed with a 3 mm slice thickness. The CT images were exported in DICOM format and imported into the RayStation planning system for subsequent analysis.

All patients in this study underwent free-breathing CT scans on the Philips Brilliance large-bore CT simulator, with target delineation performed in conjunction with diagnostic radiology CT images. As the enrolled patients received conventional fractionated radiotherapy (60 Gy / 30 fractions), 4D CT was generally not used for respiratory motion tracking. However, during target delineation, tumor motion was taken into account, and appropriate expansions were applied to the clinical target volume (CTV) and planning target volume (PTV) to ensure adequate treatment coverage. Additionally, 11 out of the 63 patients underwent 4D CT scanning based on the radiation oncologist's specific assessment, and the internal target volume (ITV) technique was used for target delineation, while the remaining patients were treated with free-breathing CT scans and thermoplastic masks to limit respiratory motion, alongside patient education for stable breathing.

Target volumes and organs at risk (OARs) were delineated by experienced radiation oncologists. The target volumes included the primary tumor (GTV-T) and involved lymph nodes

(GTV-N), which together comprised the gross tumor volume (GTV). The GTV was expanded to create the clinical target volume (CTV), which was further expanded to generate the planning target volume (PTV) with an isotropic margin of 5 mm in the lateral and anterior–posterior directions and 10 mm in the superior–inferior direction, as reviewed and approved by the attending physician. OARs, including both lungs, heart, and spinal cord, were contoured. The dose constraints referred to the RTOG 0617 protocol [32] and the clinical standards established by our institution, as outlined below:

- **PTV**: $V_{60Gy} \geq 95\%$, $D_{0.03cc} < 69$ Gy (115% of prescribed dose), Paddick conformity index (PCI) > 0.8.
- **Both lungs**: $V_{30Gy} < 18\%$, $V_{20Gy} < 28\%$, $V_{5Gy} < 50\%$, $D_{mean} < 12.5$ Gy.
- **Heart**: $V_{30Gy} < 40\%$, $D_{mean} < 25$ Gy.
- **Spinal cord**: $D_{0.03cc} < 45$ Gy.

For clinical plan evaluation, the variable "$V_{xGy}$" is defined as the volume of a structure receiving a dose greater than or equal to x Gy, while "$D_{xcc}$" is defined as the minimum dose received by x cm³ of a structure. The maximum dose of a structure is typically represented by $D_{0.03cc}$. The Paddick conformity index (PCI) was calculated using the Paddick formula [33] PCI = $(TV_{PIV})^2/(TV \cdot PIV)$, where $TV_{PIV}$ represents the volume of the target encompassed by the prescription isodose line (the prescription dose of 60 Gy), TV is the target volume, and PIV is the total volume of the prescription isodose.

## Radiotherapy plan evaluation

Radiotherapy plans based on the Halcyon accelerator were designed for all patients in the RayStation 9A planning system. To protect normal lung tissue, beam arrangements consisted of three pairs of clockwise rotational partial arcs and their corresponding counterclockwise counterparts (182° ∼ 230°, 300°∼ 60°, 130° ∼ 178°), as shown in Fig 3. Dose calculations used 6 MV flattening filter-free (FFF) beams with a RayStation Monte Carlo algorithm at 0.2% uncertainty and a dose calculation grid of 2.5 mm × 2.5 mm × 2.5 mm. Plan optimization employed the DMPO algorithm within the RayStation system, incorporating

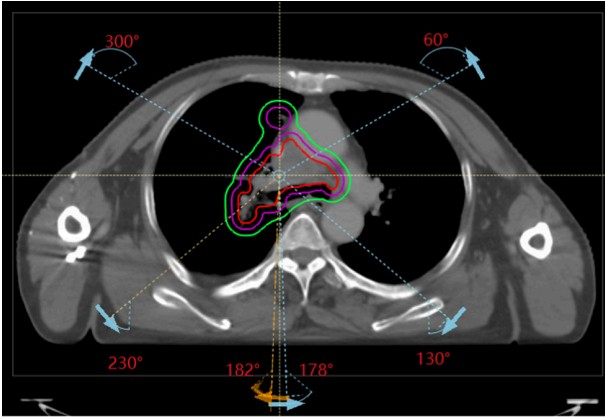

**Fig 3. Beam arrangements of VMAT plans consisted of three pairs of clockwise rotational partial arcs and their corresponding counterclockwise counterparts (182° ∼ 230°, 300°∼ 60°, 130° ∼ 178°).**

a progressive automatic optimization strategy [34] to ensure sufficient target dose coverage while minimizing normal organ dose exposure for each patient. Radiotherapy plans for each patient were optimized and calculated using the Monte Carlo algorithm in the RayStation planning system. These plans were normalized to ensure that 95% of the planning target volume (PTV) received the prescription dose of 60 Gy. The final dose was calculated using the Monte Carlo algorithm in RayStation with a statistical uncertainty of 0.2% per plan and a dose grid size of 2.5 mm. The average computation time for a typical VMAT lung plan was approximately 20–30 seconds on a GPU workstation equipped with an NVIDIA RTX A6000 (24 GB), demonstrating the clinical feasibility of GPU-accelerated Monte Carlo dose calculation.

The Halcyon radiotherapy plans (referred to as haMC) were transferred to the Eclipse planning system for recalculation using the AXB and AAA algorithms (haAXB and haAAA). In this study, the prefix "ha" is used only as a dataset label to indicate that the plans were generated for the Halcyon accelerator, whereas the underlying algorithms are Monte Carlo (MC), Acuros XB (AXB), and Anisotropic Analytical Algorithm (AAA). Clinical dose recalculations retained the same field monitor units (MUs) and plan parameters as the original design, with a consistent dose grid resolution of 2.5 mm. For the AXB algorithm, the default reporting mode was dose-to-medium, reflecting the algorithm's approach to dose computation, which calculates energy deposition in the actual medium rather than water. This choice ensures high accuracy in modeling the effects of heterogeneous tissues. Evaluation metrics included $D_2\%$, $D_{98}\%$, $D_{50}\%$, and $D_{mean}$ for the target, conformity index (PCI), homogeneity index (HI), and gradient index (GI). Dose metrics for organs at risk, including both lungs, the heart, and the spinal cord, were also assessed. DVHs and dose parameters for the haMC dataset were extracted from RayStation, while those for the haAXB and haAAA datasets were extracted from Eclipse.

The homogeneity index (HI) was calculated as: $HI = (D_{2\%} - D_{98\%})/D_{50\%}$, following the definition recommended in ICRU Report 83 [35]. The gradient index (GI) was defined as the ratio of the 50% prescription dose volume (30 Gy) to the 100% prescription dose volume (60 Gy), evaluating dose falloff outside the target [36]. Additionally, the normal tissue integral dose (NTID), calculated as the product of mean dose and the volume of normal tissue outside the PTV, was used to evaluate dose exposure to normal tissues [37].

## Statistical analysis

Statistical analysis was performed using the SciPy library. Continuous variables are expressed as mean ± standard deviation (Mean ± SD). Relative differences were calculated with haMC as the reference algorithm for all comparisons. Group comparisons were conducted using paired t-tests for data meeting the assumption of normality or Wilcoxon signed-rank tests for non-normally distributed data. Normality was assessed using the Shapiro-Wilk test. To account for potential errors arising from multiple comparisons, we incorporated a multiple testing correction in the revised manuscript. Specifically, we compared the haMC vs. haAXB and haMC vs. haAAA groups, applying the Bonferroni correction. Since two comparisons were made, the significance threshold was adjusted to $p = 0.025$ instead of 0.05. The analysis revealed that most evaluated metrics did not meet the normality assumption, with the exceptions being $PTV_{D95\%}$ and the lung dose metrics $V_{30Gy}$, $V_{20Gy}$, and $D_{mean}$. Consequently, non-parametric statistical methods were deemed more appropriate for the majority of the analyses to ensure robustness and reliability in interpreting the results.

## Results

### Cylindrical water phantom test

This study simulated dose delivery in a cylindrical water phantom using three algorithms: MC, AXB, and AAA. Absolute dose evaluation was performed using the Gamma analysis method with a 10% dose threshold, with 2% dose deviation and 2 mm distance to agreement (DTA). Results are summarized in Table 1.

The results demonstrate that MC and AXB exhibited excellent agreement, achieving high gamma pass rates under both global and local criteria. For global gamma (2%/2 mm), AXB and AAA yielded a slightly higher pass rate (100%) compared to MC and AXB (99.57%). This difference is attributable to the less stringent nature of global gamma, which is less sensitive to small discrepancies in high-gradient regions. In local gamma analysis, which better reflects boundary dose variations, MC and AXB achieved a pass rate of 99.03%, whereas MC and AAA dropped to 87.87%, confirming that AAA tends to overestimate dose in heterogeneous boundary regions. Notably, both the global and local gamma pass rates between AXB and AAA were very high (100% and 95.07%, respectively). A lateral dose profile curve at the central dose point is presented in Fig 4 to further illustrate these differences.

Table 1. **Gamma analysis for MC, AXB and AAA.** Gamma pass rate for cylindrical water phantom test. Absolute dose, 10% dose threshold, 2mm DTA and 2% dose derivation. The prefix "ha" is a dataset label specifying that the plans were generated for the Halcyon accelerator.

| Gamma Pass Rate | haMC & haAXB | haMC & haAAA | haAXB & haAAA |
|---|---|---|---|
| global (2%/2mm) | 99.57% | 99.47% | 100% |
| local (2%/2mm) | 99.03% | 87.87% | 95.07% |

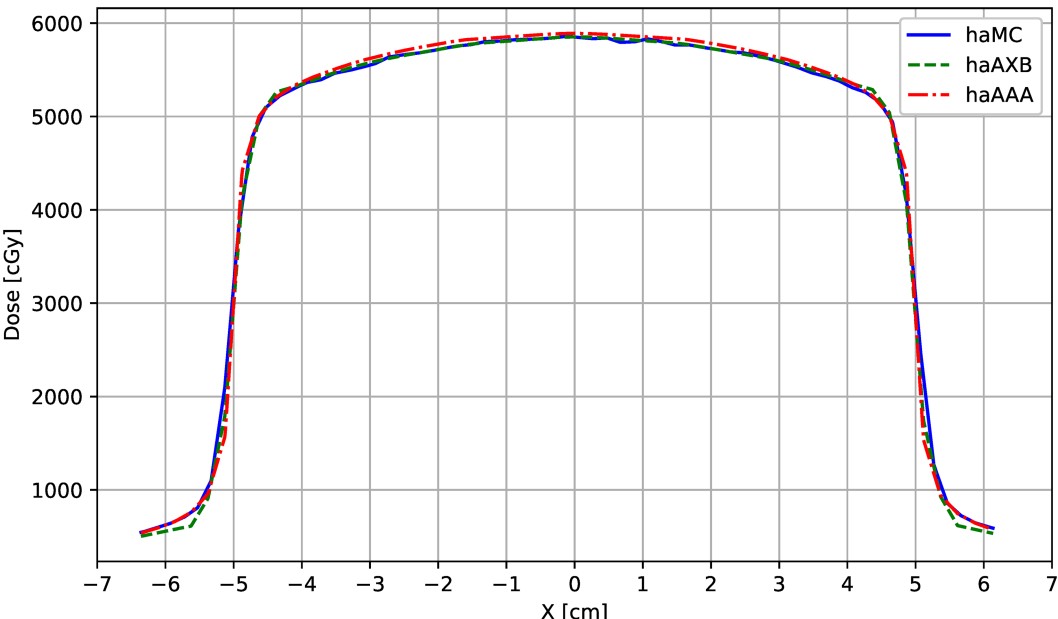

**Fig 4. Lateral dose profile curves at the central dose point for the three dose engines (MC, AXB, and AAA; shown here as haMC, haAXB, and haAAA to indicate Halcyon-specific datasets).** The curves correspond to the "Line dose" shown in Figure 1 and illustrate the differences in dose distribution across the treatment field, with dose values represented in cGy along the vertical axis and position (X) in cm along the horizontal axis.

Dose distributions were further analyzed within spherical regions centered at the intersection of the field central axis and the cylindrical phantom's central line, with radii of 3 cm and 5 cm. Dose metrics in Table 2 and the dose-volume histogram (DVH) comparison (Fig 5) revealed subtle differences in dose distributions among the three algorithms within these spherical regions. Although the variations were minimal, they reflect differences in algorithmic handling of dose deposition in homogenous water-equivalent materials, particularly under local dose evaluation conditions.

For the sphere with a 3 cm radius, the dose-volume histogram (DVH) curves of haMC and haAXB were nearly identical, reflecting excellent consistency in dose coverage between the two algorithms. However, slight deviations were observed when comparing haMC to haAAA.

**Table 2. Dose metrics for sphere with radii of 3 cm and 5 cm in the cylindrical water phantom test.** Diff(%)[1] refers to the percentage difference between group haMC and haAXB, calculated as $(haAXB - haMC)/haMC \times 100\%$; Diff(%)[2] refers to the percentage difference between group haMC and haAAA, calculated as $(haAAA - haMC)/haMC \times 100\%$.

| Dose (cGy) | Sphere (3 cm) | | | | | Sphere (5 cm) | | | | |
|---|---|---|---|---|---|---|---|---|---|---|
| | haMC | haAXB | Diff(%)[1] | haAAA | Diff(%)[2] | haMC | haAXB | Diff(%)[1] | haAAA | Diff(%)[2] |
| $D_{0.03cc}$ | 7045.5 | 7029.5 | -0.23 | 7058.5 | 0.18 | 7988 | 7971 | -0.21 | 7999.5 | 0.14 |
| $D_{2\%}$ | 6791 | 6777.5 | -0.20 | 6808.5 | 0.26 | 7457.5 | 7437.5 | -0.27 | 7473 | 0.21 |
| $D_{50\%}$ | 5676 | 5674.5 | -0.03 | 5723.5 | 0.84 | 5405 | 5413 | 0.15 | 5438.5 | 0.62 |
| $D_{mean}$ | 5728.2 | 5723.5 | -0.08 | 5768.9 | 0.71 | 5541.9 | 5544.6 | 0.05 | 5577.6 | 0.64 |
| $D_{95\%}$ | 4998.5 | 4990 | -0.17 | 5036 | 0.75 | 4400.5 | 4413 | 0.28 | 4455 | -1.24 |
| $D_{98\%}$ | 4914 | 4913 | -0.02 | 4960.5 | 0.95 | 4311 | 4324.5 | 0.31 | 4373.5 | 1.45 |

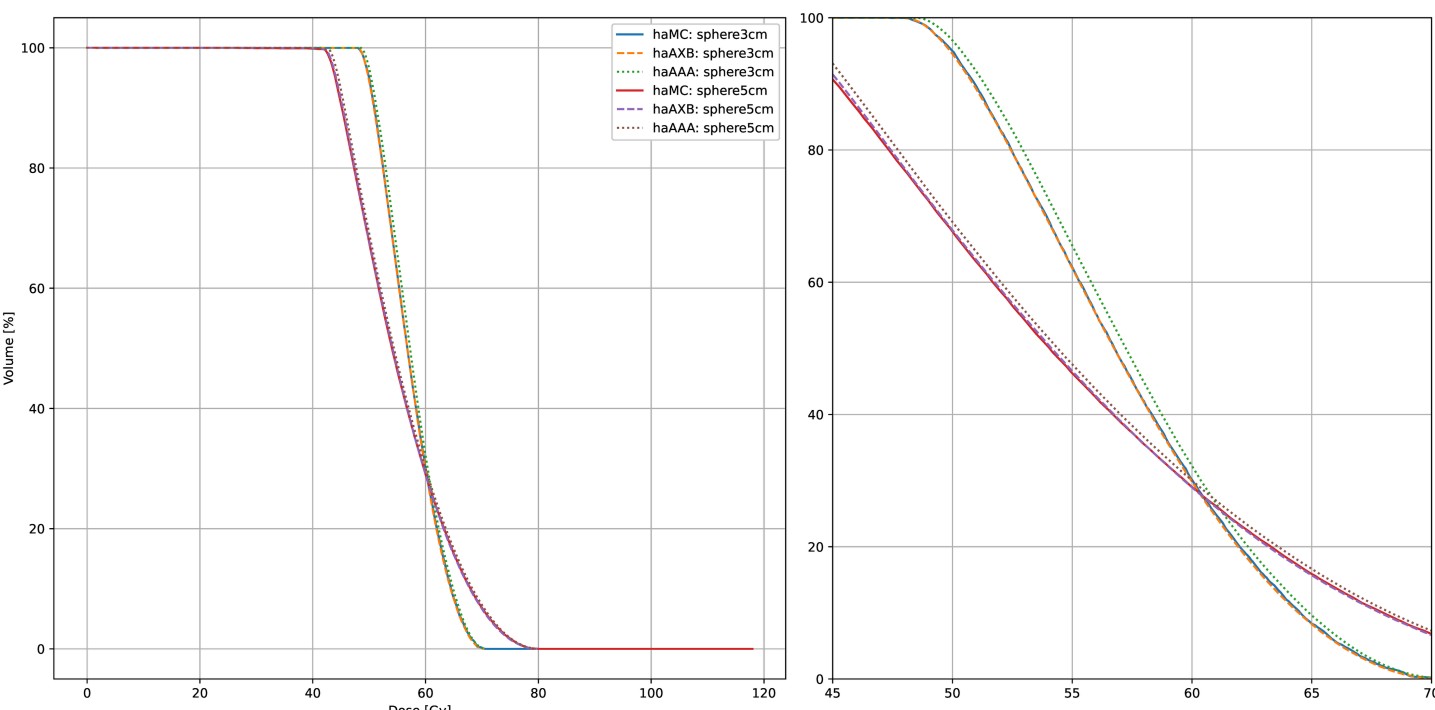

**Fig 5. DVH for spheres in cylindrical water phantom.** Dose-volume histogram (DVH) curves of haMC, haAXB, and haAAA for spheres with radii of 3 cm and 5 cm in a cylindrical water phantom with a 10x10 cm² field size at 300 MU. The right panel shows an enlarged view of the main figure, focusing on the x-axis range from 45 to 70 Gy.

For the sphere with a 5 cm radius, these differences became more apparent, particularly in high-dose regions, where the haAAA curve was slightly higher.

## Patient Specific Clinical Target Volume (CTV) and Planning Target Volume (PTV)

Table 3 summarizes the dose metrics for the clinical target volume (CTV) and planning target volume (PTV) across the MC, AXB, and AAA algorithms (reported as haMC, haAXB, and haAAA to indicate Halcyon-specific datasets).

The analysis shows that dose metrics for CTV and PTV calculated using MC (haMC) were generally higher than those obtained with AXB and AAA (refering to as haAXB and haAAA). Key findings include:

- For **CTV $D_{2\%}$**: The dose difference between haMC and haAXB was 75.6 cGy (1.18%), while the difference with haAAA was -5.7 cGy (-0.09%). Similarly, for **$PTV_{D2\%}$**, haMC exceeded haAXB by 76.3 cGy (1.19%) and was lower than haAAA by 2.1 cGy (-0.03%). This suggests closer agreement between haMC and haAAA in high-dose regions of CTV and PTV.
- For **CTV $D_{50\%}$** and **PTV $D_{50\%}$**: Median doses for haMC were 61.2 cGy (0.98%) and 62.8 cGy (1.01%) higher than those for haAXB, and 8.6 cGy (0.14%) and 28.8 cGy (0.46%) higher than those for haAAA, respectively. Mean dose metrics (**CTV $D_{mean}$**) followed a similar trend.
- For **CTV $D_{95\%}$** and **CTV $D_{98\%}$**: For minimum dose metrics, haMC was 51 cGy (0.84%) and 17.9 cGy (0.31%) higher than haAXB for CTV, and 50.4 cGy (0.83%) and 65 cGy (1.08%) higher than haAAA. For **PTV**, haMC exceeded haAXB by 53.8 cGy (0.90%) and 17.9 cGy (0.31%) for **$D_{95\%}$** and **$D_{98\%}$**, respectively, with larger differences of 105.2 cGy (1.78%) and 72.8 cGy (1.27%) compared to haAAA.

Figs 6 and 7 illustrate statistical boxplots summarizing the results for 63 patients across the haMC, haAXB, and haAAA groups. Statistical analysis revealed significant differences (p < 0.025) in target dose distribution and coverage among the three algorithms. Nevertheless, the absolute differences between haMC and haAXB or haAAA remained within 1% of

**Table 3. Statistical Comparison of Dose Metrics for Clinical Target Volume (CTV) and Planning Target Volume (PTV).** Columns 2, 3, and 6 display the mean and standard deviation (SD) values for the 63 enrolled patients. The fourth column presents the difference between haAXB and haMC (haAXB-haMC), with the corresponding paired test p-value shown in the fifth column. Similarly, the seventh and eighth columns show the difference between haAAA and haMC (haAAA-haMC), along with their paired test p-values. "n.s." indicates that the difference between groups was not statistically significant (p ≥ 0.025).

| Dose Metrics (cGy) | haMC | haAXB | Diff | p value | haAAA | Diff | p value |
|---|---|---|---|---|---|---|---|
| CTV $D_{2\%}$ | 6491.3±57.8 | 6415.7±59.4 | -75.6 | <0.001 | 6497.0±77.2 | 5.7 | n.s. |
| CTV $D_{50\%}$ | 6306.4±39.5 | 6245.2±45.3 | -61.2 | <0.001 | 6297.8±63.6 | -8.6 | n.s. |
| CTV $D_{mean}$ | 6304.6±38.1 | 6243.3±44.4 | -61.3 | <0.001 | 6291.3±61.2 | -13.3 | n.s. |
| CTV $D_{95\%}$ | 6148.9±25.1 | 6097.9±37.5 | -51.0 | <0.001 | 6098.5±54.5 | -50.4 | <0.001 |
| CTV $D_{98\%}$ | 6110.4±26.0 | 6061.6±39.7 | -48.8 | <0.001 | 6045.4±56.3 | -65.0 | <0.001 |
| PTV $D_{2\%}$ | 6486.4±56.0 | 6410.1±57.3 | -76.3 | <0.001 | 6488.5±78.9 | 2.1 | n.s. |
| PTV $D_{50\%}$ | 6272.2±31.5 | 6209.4±39.8 | -62.8 | <0.001 | 6243.4±58.9 | -28.8 | <0.001 |
| PTV $D_{mean}$ | 6250.1±28.6 | 6188.7±38.0 | -61.4 | <0.001 | 6214.1±53.3 | -36.0 | <0.001 |
| PTV $D_{95\%}$ | 6000.0±0.0 | 5946.2±30.9 | -53.8 | <0.001 | 5894.8±47.1 | -105.2 | <0.001 |
| PTV $D_{98\%}$ | 5818.5±39.2 | 5800.6±54.5 | -17.9 | 0.001 | 5745.7±63.3 | -72.8 | <0.001 |
| PTV $D_{0.03cc}$ | 6630.2±68.9 | 6556.3±68.6 | -73.9 | <0.001 | 6647.5±157.5 | 17.3 | n.s. |

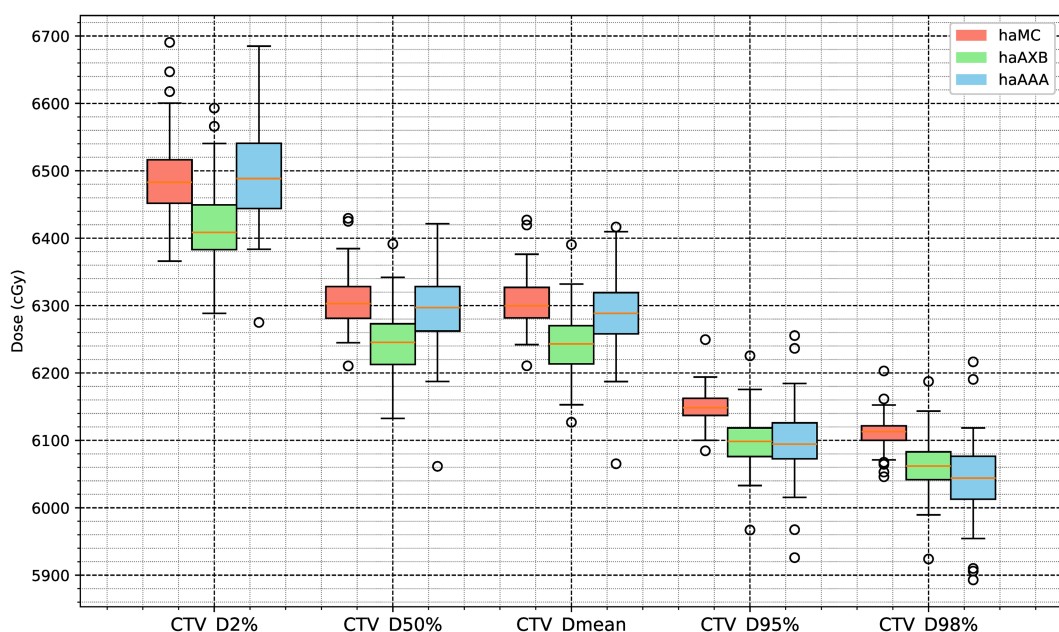

**Fig 6. Boxplot for CTV.** Boxplot comparison of dose metrics for clinical target volume (CTV) across the three algorithms: haMC (red), haAXB (green), and haAAA (blue).

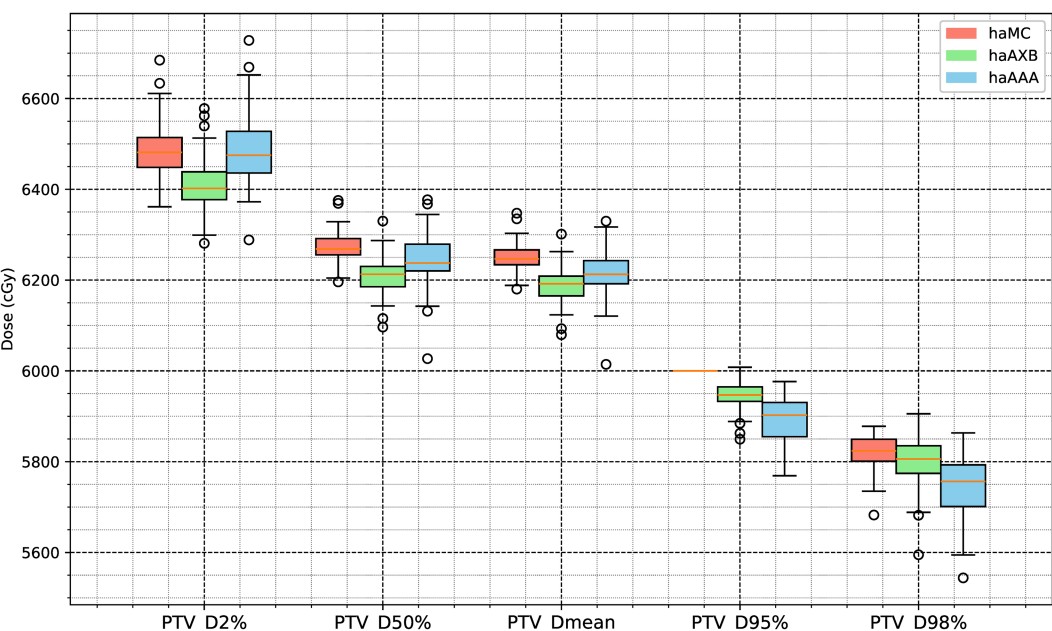

**Fig 7. Boxplot for PTV.** Boxplot comparison of dose metrics for planning target volume (PTV) across the three algorithms: haMC (red), haAXB (green), and haAAA (blue).

the prescription dose, indicating a general similarity in clinical target dose coverage across the algorithms.

Figs 8 and 9 also display the average dose-volume histogram (DVH) curves for the 63 patients, comparing haMC, haAXB, and haAAA groups. The DVH curves highlight distinct patterns: in high-dose regions of the target (e.g., D2%), haMC results closely aligned with haAAA, whereas in low-dose regions (e.g., D95%), the dose distribution of haMC was more similar to that of haAXB. When using PTV D95% as a reference for equivalent prescription dose, haAXB demonstrated greater agreement with haMC in terms of dose coverage than haAAA. These observations provide important insights for clinical decision-making, particularly in selecting and verifying algorithms for radiotherapy dose calculations.

In evaluating the quality of target area dose distributions, the Conformity Index (PTV$_{CI60Gy}$), Homogeneity Index (HI), and Gradient Index (GI) are key indicators for assessing radiotherapy plan quality:

- **Conformity Index (PTV$_{CI60Gy}$)**: The PTV$_{CI60Gy}$ for the haMC group was $0.850 \pm 0.037$, slightly higher than the haAXB group ($0.848 \pm 0.037$), and substantially higher than the haAAA group ($0.796 \pm 0.054$). Statistical analysis revealed a significant difference between haMC and haAAA ($p < 0.001$), while no significant difference was found between haMC and haAXB ($p > 0.025$). This suggests that dose calculation using haAXB on plans optimized with haMC does not seem to show a significant degradation in target conformity. Furthermore, the similarity in PTV D95% and D98% dose values between haMC and haAXB supports their comparable target conformity.
- **Homogeneity Index (HI)**: For the homogeneity index, the HI for haMC was $0.106 \pm 0.012$, significantly higher than that of the haAXB group ($0.098 \pm 0.011$), and significantly lower than that of the haAAA group ($0.119 \pm 0.014$) ($p < 0.001$). Despite significant statistical

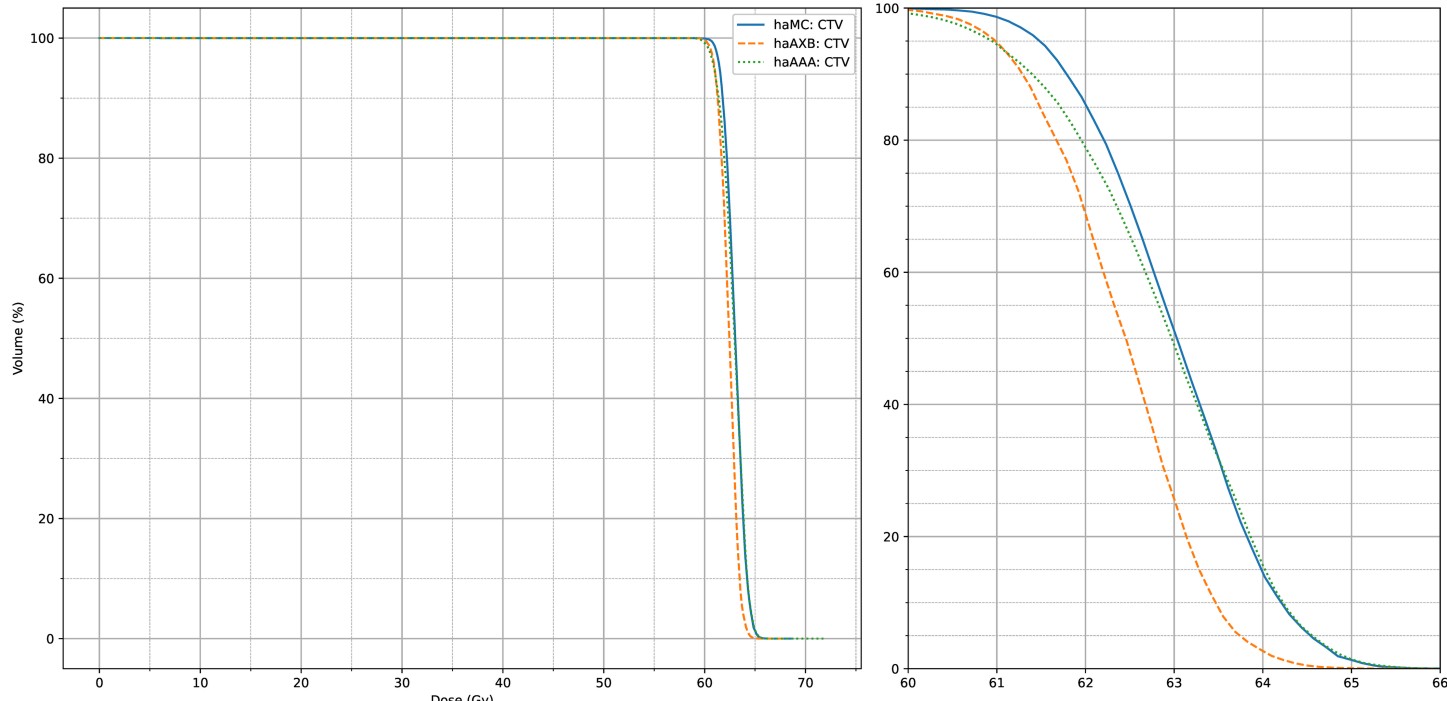

**Fig 8. Average DVH for CTV.** The average CTV dose-volume histogram (DVH) curves of haMC, haAXB and haAAA for 63 patients. The right panel shows an enlarged view of the main figure, focusing on the x-axis range from 60 to 66 Gy.

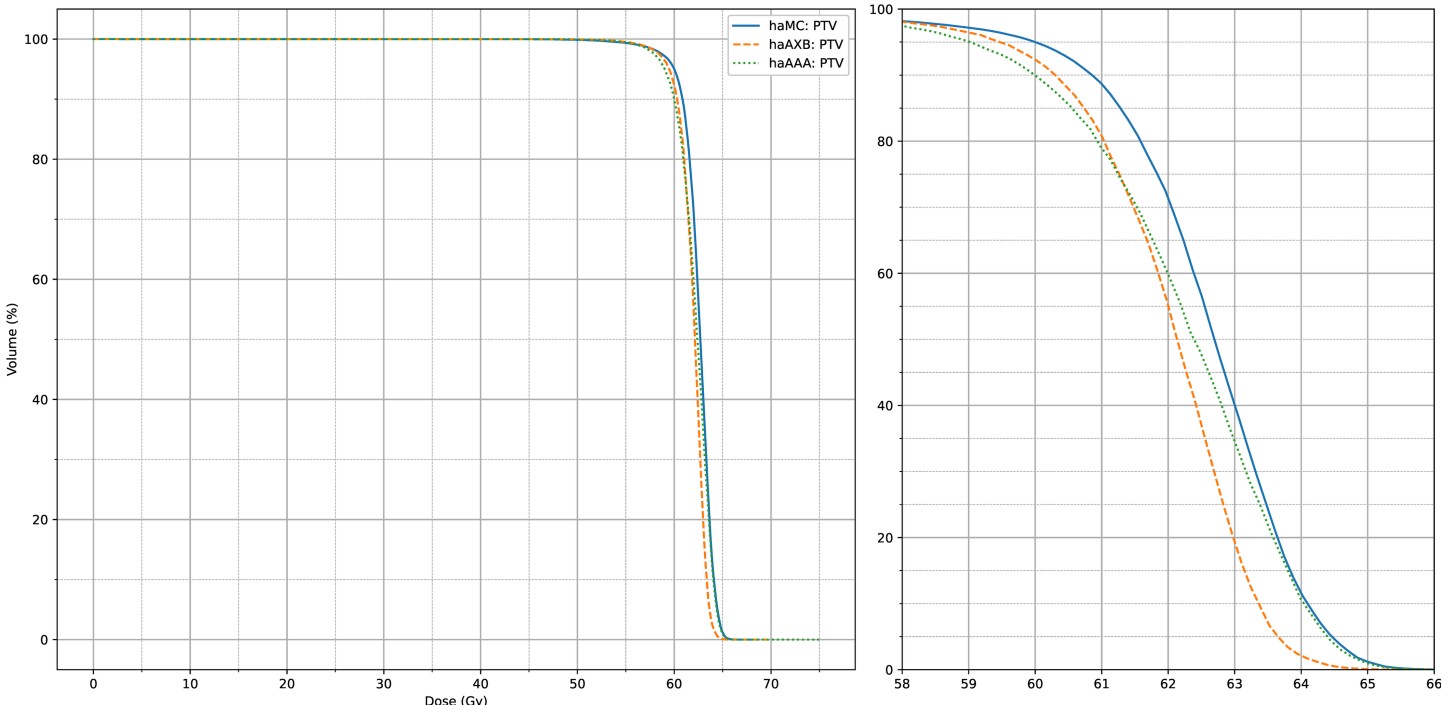

**Fig 9. Average DVH for PTV.** The average PTV dose-volume histogram (DVH) curves of haMC, haAXB and haAAA for 63 patients. The right panel shows an enlarged view of the main figure, focusing on the x-axis range from 58 to 66 Gy.

differences between the groups, the absolute differences in the homogeneity index were minimal (approximately 0.01).

- **Gradient Index (GI)**: The GI for the haMC group was 3.78 ± 0.67, significantly lower than the haAXB group (4.00 ± 0.80) and the haAAA group (4.08 ± 0.97). This indicates that the use of haAXB and haAAA to recalculate the radiotherapy plans generated by haMC leads to some greater value in the gradient index, with absolute changes ranging from 0.2 to 0.3.
- **Normal tissue integral dose (NTID)**: The haMC group had an NTID of 112800 ± 36718 $Gy \cdot cm^3$, which was significantly higher than the haAXB group (110510 ± 36075 $Gy \cdot cm^3$), with no significant difference between haMC and the haAAA group (112740 ± 36725 $Gy \cdot cm^3$).

The distributions of PCI, HI, GI, and NTID for the three algorithms are illustrated in Fig 10, which complements the numerical results by visualizing their variability. As shown above, haMC and haAXB exhibited comparable conformity indices, and both groups showed high consistency in PTV $D_{95\%}$ dose coverage. Therefore, in clinical evaluations of intensity-modulated radiotherapy plans, plans recalculated using haAXB based on haMC demonstrate relatively better clinical satisfaction in terms of target area dose coverage and conformity. This indicates that recalculated plans using haAXB remain feasible for meeting target coverage and protecting normal tissues outside the target area. However, since the plan optimization and dose calculation in this study were based on haMC, differences remain when evaluating target homogeneity and gradient indices between haMC, haAXB, and haAAA.

Differences in doses to organs at risk (OARs) were observed among the three calculation algorithms (MC, AXB, and AAA), as summarized in Table 4.

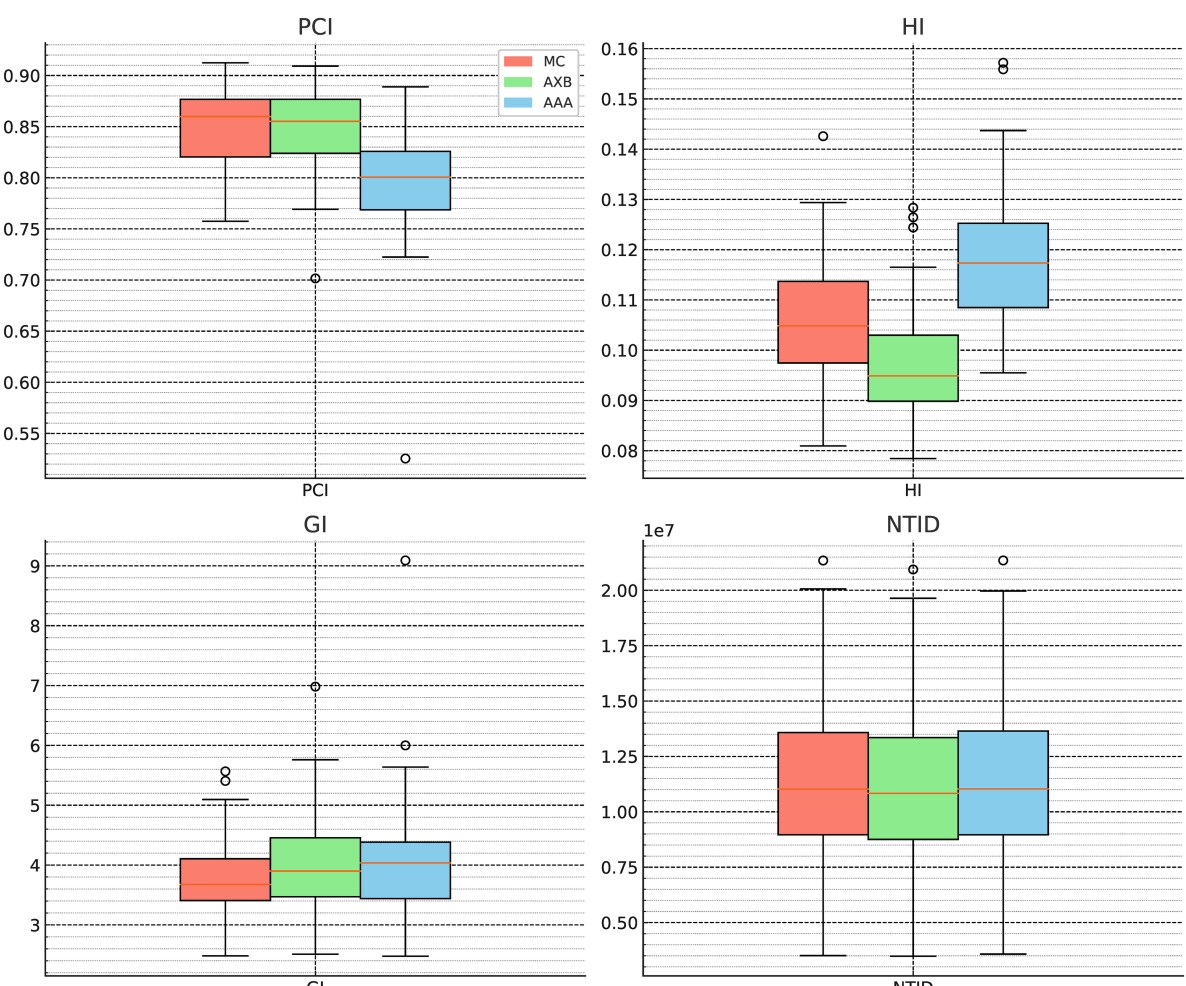

**Fig 10. Box-and-whisker plots showing the distributions of (a)Paddick Conformity Index (PCI), (b) Homogeneity Index (HI), (c) Gradient Index (GI), and (d) Normal Tissue Integral Dose (NTID) for the three algorithms (MC: salmon, AXB: light green, AAA: sky blue).** These plots complement the tabulated values by illustrating the variability and relative differences across the algorithms. The "ha" prefix in haMC, haAXB, and haAAA labels (used in tables) denotes Halcyon-specific datasets.

**Table 4**. **Statistical Comparison of Dose Metrics for organs at risk (OAR).** The fourth column presents the difference between haAXB and haMC (haAXB-haMC), with the corresponding paired test p-value shown in the fifth column. Similarly, the seventh and eighth columns show the difference between haAAA and haMC (haAAA-haMC), along with their paired test p-values.

| Dose Metrics | haMC | haAXB | Diff | p value | haAAA | Diff | p value |
|---|---|---|---|---|---|---|---|
| Lungs $D_{mean}$(cGy) | 1030.0±257.8 | 1001.1±251.8 | -28.9 | <0.001 | 1018.4±254.1 | -11.6 | <0.001 |
| Heart $D_{mean}$(cGy) | 739.0±444.1 | 702.4±434.8 | -36.5 | <0.001 | 715.8±443.7 | -23.2 | <0.001 |
| SpinalCord $D_{0.03cc}$(cGy) | 3352.3±658.9 | 3282.8±651.3 | -69.5 | <0.001 | 3324.1±659.0 | -28.2 | <0.001 |
| Lungs $V_{5Gy}$(%) | 36.3±8.2 | 35.4±8.0 | -0.9 | <0.001 | 36.5±8.4 | 0.15 | 0.016 |
| Lungs $V_{10Gy}$(%) | 28.1±6.8 | 27.5±6.7 | -0.5 | <0.001 | 27.8±6.8 | -0.25 | <0.001 |
| Lungs $V_{20Gy}$(%) | 18.5±5.6 | 18.0±5.5 | -0.5 | <0.001 | 18.1±5.6 | -0.40 | <0.001 |
| Lungs $V_{30Gy}$(%) | 12.3±4.5 | 11.9±4.4 | -0.4 | <0.001 | 11.9±4.4 | -0.39 | <0.001 |
| Heart $V_{30Gy}$(%) | 7.4±6.1 | 7.0±5.9 | -0.5 | <0.001 | 7.2±6.0 | -0.29 | <0.001 |

- **Lungs $D_{mean}$**: The mean lung dose calculated using haMC was 1030.0 ± 257.76 cGy, which was higher than the doses calculated using haAXB (1001.1 ± 251.8 cGy) and haAAA (1018.4 ± 254.0 cGy).
- **Heart $D_{mean}$**: Similarly, the mean heart dose in the haMC group was 739.01 ± 444.1 cGy, exceeding the values from haAXB (702.45 ± 434.8 cGy) and haAAA (715.8 ± 443.7 cGy).
- **Spinal Cord $D_{0.03cc}$**: The maximum dose to 0.03 cc of the spinal cord was highest in the haMC group (3352.3 ± 658.89 cGy), compared to haAXB (3282.8 ± 651.3 cGy) and haAAA (3324.1 ± 659.1 cGy). These differences are visualized in Fig 11.

For the lung volume receiving 5 Gy ($V_{5Gy}$), haMC produced values 0.91% higher than haAXB and 0.15% lower than haAAA, showing closer agreement with haAAA. At higher dose thresholds, the differences between haMC and haAXB were 0.52%, 0.49%, and 0.39% for lung V10Gy, V20Gy, and V30Gy, respectively, while the differences between haMC and haAAA were 0.26%, 0.39%, and 0.39%. For the heart volume receiving 30 Gy (V30Gy), haMC differed from haAXB by 0.47% and from haAAA by 0.29%. These findings are further detailed in Figs 11 and 12, as well as the average dose-volume histogram of 63 patients in Fig 13.

In dose metrics of clinical importance for lung and heart protection, the differences between haMC and the other two groups (haAXB and haAAA) were generally within 0.5%. For the mean doses to the lungs and heart, the differences among the three algorithms were within 30 cGy, corresponding to relative differences of less than 3% (based on a reference mean dose of 1000 cGy). These results highlight the consistency of the three algorithms in safeguarding organs at risk, with differences remaining within clinically acceptable ranges.

Histograms for two commonly used clinical target dose metrics, PTV $D_{95\%}$ and $D_{50\%}$, as well as two key normal lung dose metrics, lungs $D_{mean}$ and $V_{20Gy}$, are presented in Figs 14 and 15, respectively. These histograms illustrate the variations in PTV and lung dose metrics

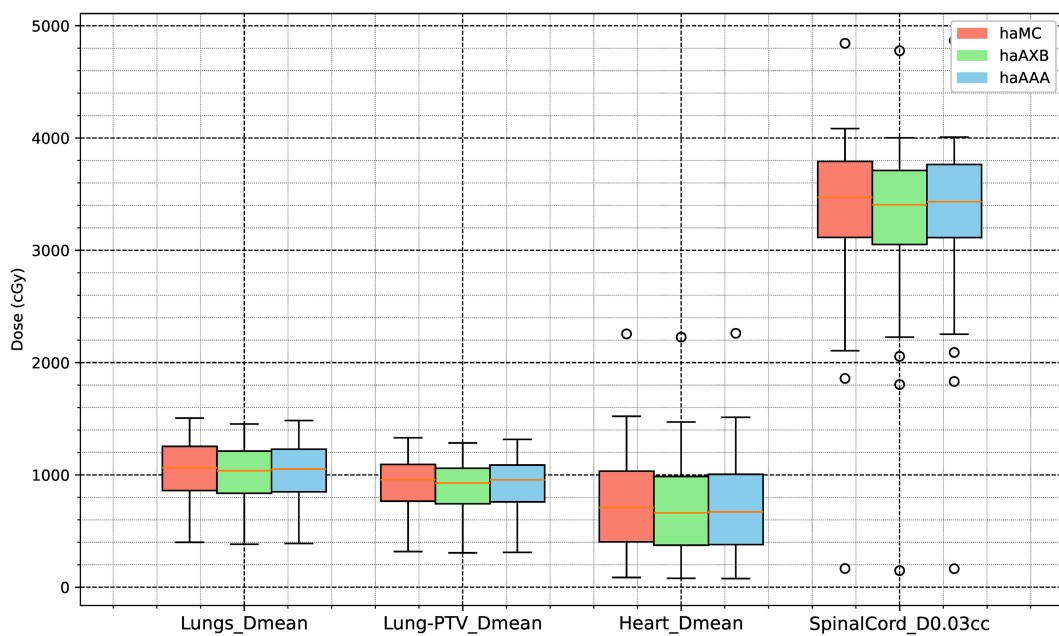

**Fig 11. Boxplot comparison of mean dose metrics for organs at risk (OARs) across the three groups: haMC (red), haAXB (green), and haAAA (blue).**

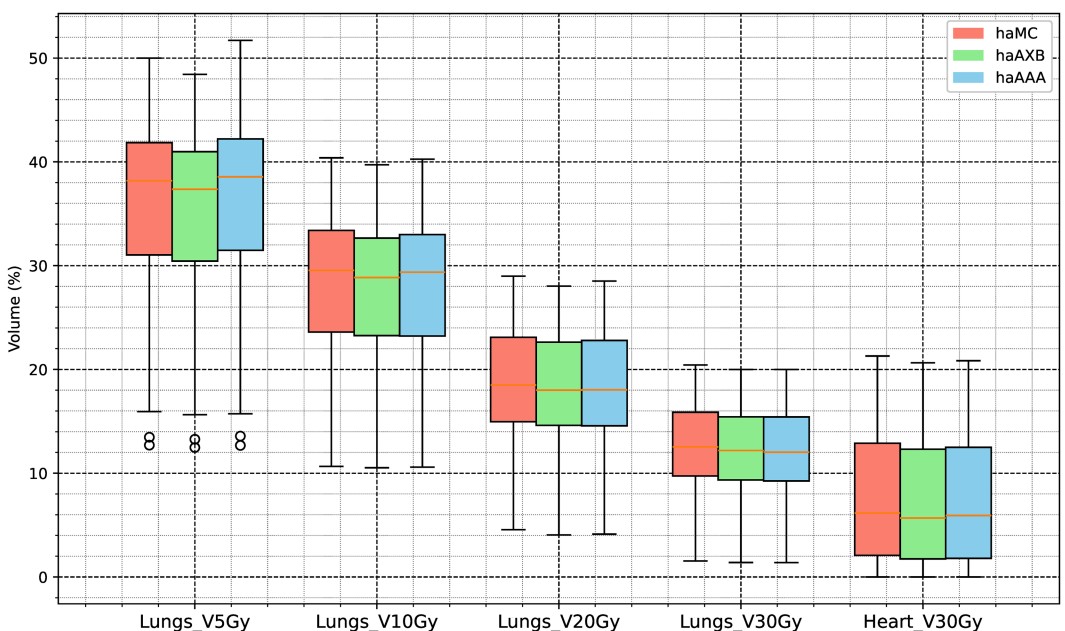

**Fig 12. Boxplot comparison of volume-based dose metrics for the lungs and heart across the three algorithms: haMC (red), haAXB (green), and haAAA (blue).**

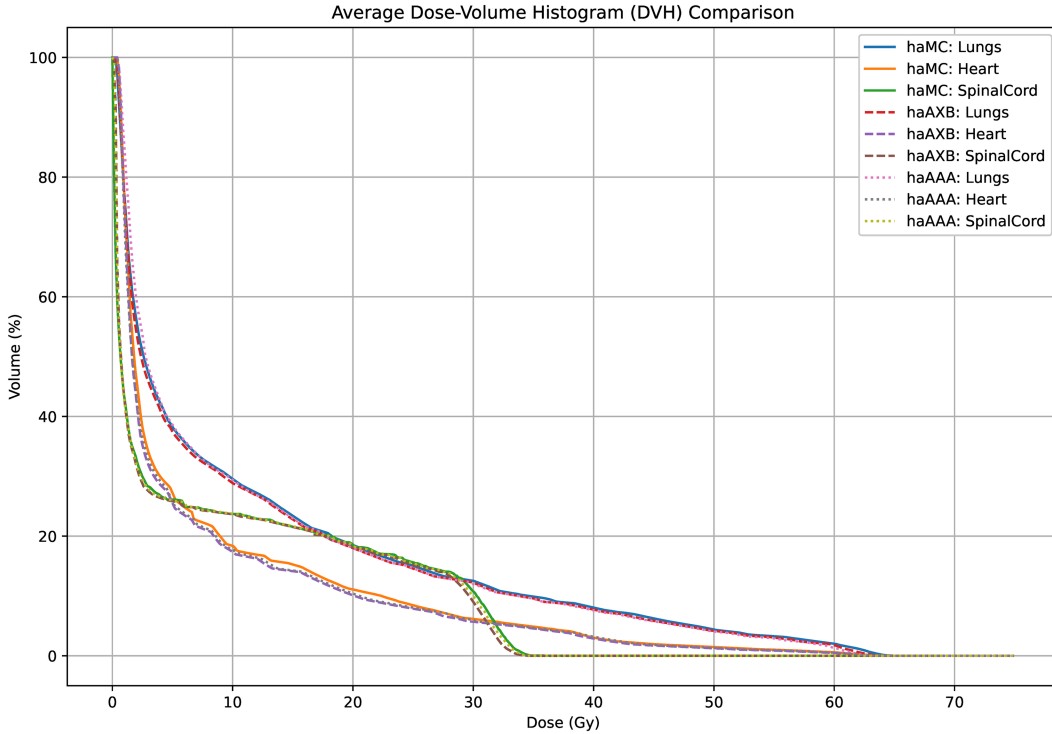

**Fig 13. Average (63 patients) dose-volume histogram (DVH) comparison for the lungs, heart, and spinal cord across the three algorithms: haMC (solid lines), haAXB (dashed lines), and haAAA (dotted lines).**

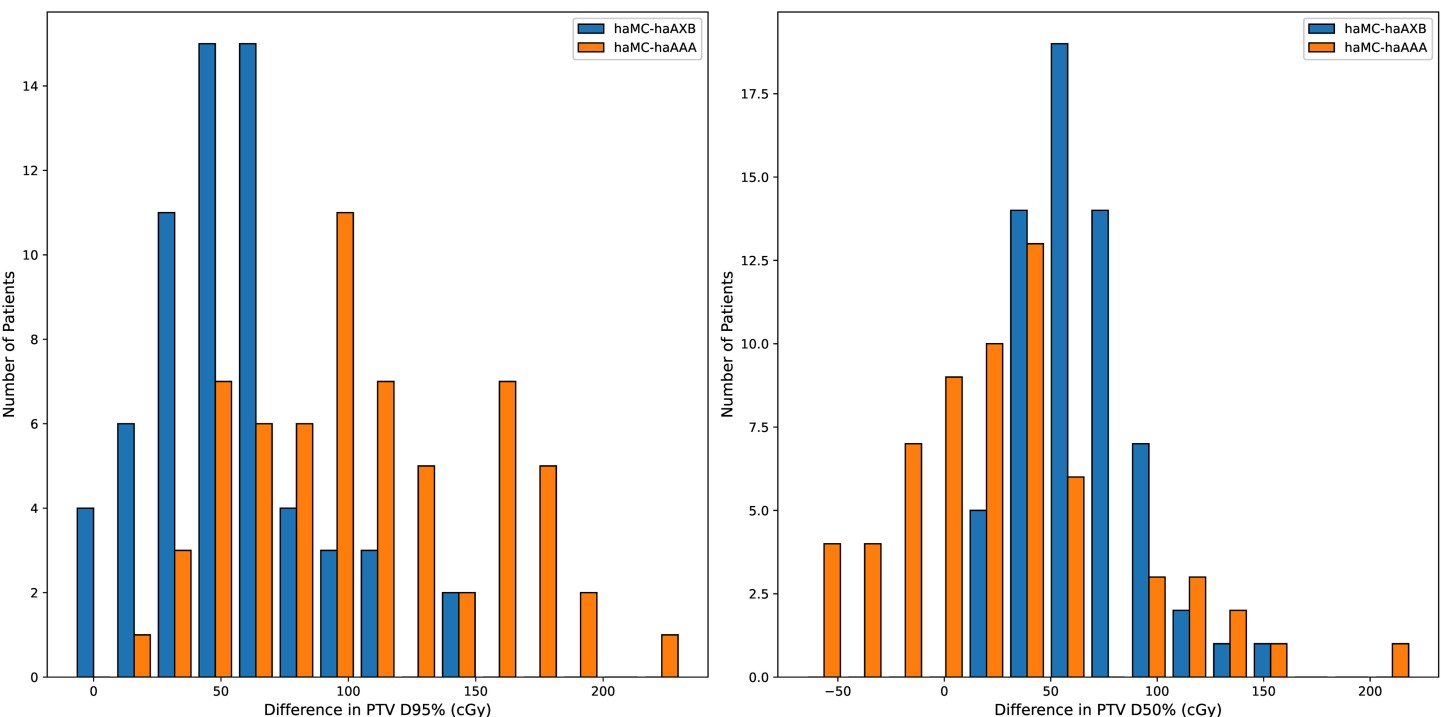

**Fig 14. Histograms of dose differences in PTV dose metrics, $D_{95\%}$ (left) and $D_{50\%}$ (right), across different algorithms.** The vertical axis represents the number of patients within each dose difference interval. For $D_{95\%}$, haMC and haAXB show closer agreement (blue bars are concentrated near zero).

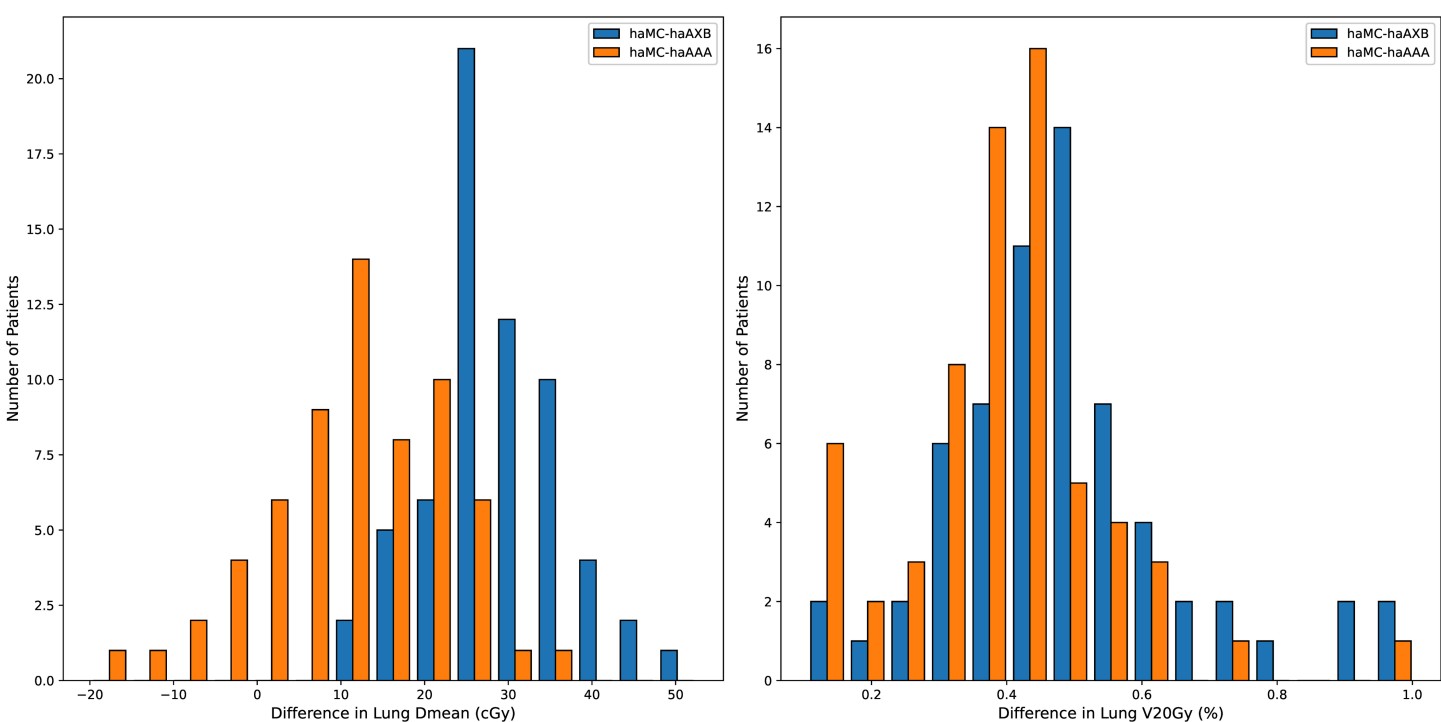

**Fig 15. Histograms of dose differences in normal lung dose metrics, Lung $D_{mean}$ and $V_{20Gy}$ (right), across different algorithms.** The vertical axis represents the number of patients within each dose difference interval. For both metrics, haAAA shows closer agreement with haMC (orange bars more centered), while haAXB generally reports lower values (blue bars shifted rightward).

across different algorithms for each patient, providing a clearer visualization of the algorithmic differences in actual treatment plans.

## Discussion

In this study, we examined the dose differences between RayStation Monte Carlo (RMC) and Eclipse Acuros XB (AXB) and AAA algorithms within the context of a clinically realistic Halcyon workflow. In our institution, RayStation is the primary TPS used for plan optimization and clinical reporting, while Eclipse ARIA functions as the record-and-verify (R&V) platform. Consequently, treatment plans generated in RayStation are recalculated in Eclipse using AXB or AAA for approval without renormalization, thereby preserving the original monitor units (MUs) and plan parameters documented in the RayStation-approved plan report. This approach ensures that the recalculated Eclipse plan used for delivery remains consistent with the clinically validated RayStation plan, in line with established cross-TPS commissioning practices [25].

Using a cylindrical water phantom and a large cohort of 63 conventionally fractionated lung cancer patients, our analysis demonstrated that RMC and AXB exhibit excellent agreement in target dose metrics, with differences typically within 1%. AAA showed slightly larger deviations, particularly in heterogeneous lung regions, but these differences remained clinically negligible. For organs at risk (OARs), RMC generally yielded marginally higher dose estimates than AXB and AAA; however, these variations did not exceed clinically acceptable limits. We also observed that for some low-dose parameters such as NTID and certain OAR doses (typically around 10–20% of the prescription dose), AAA values appeared numerically closer to RMC. This effect likely reflects the greater influence of dose smoothing in low-dose regions, and the absolute differences remained small and clinically insignificant.

Our findings are consistent with prior studies demonstrating that AXB shows better agreement with MC than AAA in heterogeneous conditions such as lung or bone [26–30]. Specifically, in radiation therapy plans for patients with lung cancer, MC and AXB demonstrated strong agreement in high-dose regions, particularly around the PTV $D_{95}$. Notably, our work extends this evidence base in several important ways. First, while Saini et al. (2021) validated Halcyon beam modeling in RayStation using phantom measurements and AXB dose comparisons, they did not investigate cross-TPS consistency in patient cases. By incorporating 63 lung cancer patients with anatomically complex targets, our study provides the first large-scale, patient-based validation of RayStation MC against Eclipse AXB/AAA in a true dual-TPS Halcyon workflow. Second, we confirmed that these algorithmic differences, although statistically significant, are of limited clinical relevance (<1% for target coverage and minimal OAR impact), supporting the safe and efficient interchangeability of MC and AXB for clinical verification.

Importantly, our workflow preserves RayStation-based prescription reporting while ensuring delivery compatibility with Eclipse ARIA. Although minor reductions in PTV $D_{95}$ may be observed after Eclipse recalculation, Halcyon's daily CBCT-guided IGRT ensures that CTV coverage is maintained with submillimeter setup accuracy, mitigating any practical dosimetric impact. For institutions that design, evaluate, and document plans in Eclipse, a slight renormalization after AXB recalculation may be applied to align PTV coverage for reporting purposes; however, our approach avoids modifying the MU or prescription parameters of the clinically approved plan and is therefore preferred for cross-TPS verification. Furthermore, our results confirm that RayStation's MC algorithm, which has been benchmarked against

reference-grade Monte Carlo codes such as DOSXYZnrc and TOPAS [38,39], provides dosimetric accuracy within 2–3% for heterogeneous cases, meeting international standards such as IAEA TRS-430. This supports the validity of our use of RayStation MC as the reference in this study.

Previous studies have primarily applied the Halcyon model in the Varian Eclipse radiotherapy planning system. Saini et al. (2021) [25] modeled Halcyon in RayStation 9B and performed a comparative analysis of 15 VMAT and IMRT test plans across different anatomical regions. They reported an average dose difference of $0.0 \pm 1.1\%$ in water phantoms. Numerous studies have also investigated dose calculation algorithms, examining differences in dose distributions, such as between AAA and AXB or CC and AAA/AXB. These studies commonly concluded that convolution algorithms possess certain advantages in accounting for heterogeneous densities [26,40–42]. Kroon et al. (2013) [43] highlighted that AcurosXB provides more accurate dose predictions in low-density heterogeneous regions for lung cancer VMAT plan evaluations, while AAA tends to overestimate doses compared to AcurosXB. Similarly, Padmanaban et al. (2014) [41] observed that in esophageal VMAT plans, AcurosXB predicted lower average doses for the GTV, PTV, and OARs. Overestimation by the AAA algorithm may lead to underdosage in tumors, potentially reducing tumor control probability. Studies comparing the Monte Carlo algorithm in Monaco have shown that MC algorithms achieve higher dose calculation accuracy in complex geometries [44,45]. For lung cancer patients, where significant density heterogeneity exists in lung tissues, traditional pencil beam algorithms are no longer suitable. Convolution algorithms like CC and AAA, which incorporate corrections for heterogeneous density tissues, serve as viable alternatives. However, the Monte Carlo algorithm, with its direct simulation of particle transport processes, theoretically achieves the highest accuracy.

For radiotherapy plan design on the Halcyon accelerator, our institution primarily uses RayStation for treatment planning across all linacs, including TrueBeam, Trilogy, Infinity, and Synergy. Radiation oncologists are accustomed to contouring and evaluating plans in RayStation, and maintaining the same TPS for both Halcyon and other machines ensures workflow consistency and facilitates plan transfer between different accelerators. This practice is mainly the result of our historical transition from Pinnacle to RayStation and the limited number of Eclipse licenses available, rather than an inherent technical superiority of RayStation over Eclipse. Although Eclipse is required for final dose recalculation and plan transfer to Halcyon, RayStation remains the routine platform for plan optimization and evaluation in our center, while Eclipse serves primarily as a verification and delivery interface. This dual-TPS workflow reflects our institutional context and allows efficient and consistent clinical operations. In addition, the GPU-accelerated Monte Carlo engine in RayStation allows final dose calculation within approximately 20–30 seconds per plan (dose grid size: 2.5 mm; statistical uncertainty: 0.2% per plan) on an NVIDIA RTX A6000 (24 GB) GPU, further supporting its integration into routine clinical workflows without introducing planning delays.

The findings of this study indicate that RayStation Monte Carlo algorithm (RMC) delivers dose calculations equivalent to those of AXB algorithm in Eclipse while benefiting from RayStation's efficient optimization and computation features. This approach enables consistent plan design and evaluation across different accelerators, simplifying the workflow for clinicians and physicists. Since optimization in RayStation is based on ensuring that 95% of the PTV volume receives the prescribed dose, the dose recalculated in Eclipse may result in slight variations in target coverage. These discrepancies may give the impression of a potential

reduction in target coverage. To address this, slight adjustments during dose normalization—such as prescribing 100.5% of the dose to cover 95% of the PTV in RayStation — can ensure alignment with AXB-based calculations in Eclipse. At our institution, when designing and clinically evaluating Halcyon treatment plans using RayStation MC, it remains the standard practice to use 100% of the prescription dose to cover 95% of the PTV volume. In fact, Halcyon's daily CBCT-guided radiotherapy (IGRT) offers greater positional accuracy than traditional accelerators with less CBCT frequency (usually once or twice a week). Although the recalculated dose in Eclipse may seem to show a slight reduction in PTV $D_{95}$, the daily CBCT-guided setup correction ensures that the CTV receives the prescribed dose, compensating for minor variations in PTV dose coverage. For **users at other institutions** who design, evaluate, and generate clinical reports using the Eclipse TPS and also have access to the RayStation TPS, one strategy involves creating treatment plans using MC algorithm in RayStation, transferring these plans to Eclipse for dose recalculation with the haAXB algorithm, and then normalizing the prescribed dose for evaluation and treatment reporting. This strategy not only maintains dose consistency but also provides flexible and practical clinical implementation solutions.

This study has several limitations. First, it was a single-center retrospective study with a cohort of 63 patients, which, although larger than many prior algorithm-comparison studies, may still limit the statistical power and generalizability of the findings. Future multi-center studies with larger datasets would be beneficial to confirm the reproducibility of our results across different clinical settings. Second, while this study validated the consistency between RayStation Monte Carlo algorithm and AXB and AAA algorithms in Eclipse for the Halcyon accelerator, the conclusions are currently restricted to this specific cross-TPS configuration. Whether similar results would be observed for other treatment planning systems or accelerators from different manufacturers remains to be investigated. Third, the dose–volume metrics were calculated separately in RayStation and Eclipse, which may introduce slight differences related to the internal volume computation methods of each TPS. The use of a 5-mm CT slice thickness may also contribute to minor discrepancies, partly due to potential end-capping effects in the structure volume calculation. These factors may explain a portion of the observed differences in dose distribution between RayStation and Eclipse. Future research should aim to address these limitations by incorporating multi-center data, exploring a broader range of tumor sites and treatment techniques (including SBRT and hypofractionated regimens), and investigating additional cross-platform scenarios. Such studies would provide a more comprehensive understanding of the applicability and limitations of different dose calculation algorithms and strengthen the clinical relevance of cross-TPS validation for Halcyon workflows.

## Conclusion

This study demonstrated that the dose calculations by RayStation Monte Carlo (RMC) and Acuros XB (AXB) algorithms showed strong consistency, particularly in high-dose regions around the PTV D95%. Compared to MC, AAA tended to slightly underestimate the dose near the prescription level, though all recalculated plans using AXB and AAA remained within clinically acceptable limits. MC yielded higher OAR doses than AXB and AAA, with AXB consistently reporting the lowest values. These results indicate that when Halcyon VMAT plans are designed and evaluated using RayStation Monte Carlo algorithm, subsequent dose recalculations in Eclipse can be reliably performed using Acuros XB (AXB) for clinical implementation.

## Supporting information

**S1 File. patients_data.csv.** Contains anonymized patient characteristics, including tumor location, stage, and target volume statistics.
(CSV)

**S2 File. haMC_dose.csv.** Dosimetric parameters derived from plans calculated using the Monte Carlo (haMC) algorithm in RayStation.
(CSV)

**S3 File. haAXB_dose.csv.** Dosimetric parameters from recalculated plans using the Acuros XB (haAXB) algorithm in Eclipse.
(CSV)

**S4 File. haAAA_dose.csv.** Dosimetric parameters from recalculated plans using the Anisotropic Analytical Algorithm (haAAA) in Eclipse.
(CSV)

## Author contributions

**Conceptualization:** Kainan Shao, Weijun Chen.

**Data curation:** Kainan Shao, Chaojun Chai.

**Formal analysis:** Chaojun Chai.

**Funding acquisition:** Yiwei Yang, Fenglei Du.

**Methodology:** Yiwei Yang, Weijun Chen, Fenglei Du.

**Software:** Chaojun Chai, Yiwei Yang.

**Visualization:** Chaojun Chai, Yiwei Yang.

**Writing – original draft:** Kainan Shao, Fenglei Du.

**Writing – review & editing:** Weijun Chen, Fenglei Du.

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
