## [Decision Letter · Decision Letter 0]

27 Jun 2025

PONE-D-25-21057Dosimetric comparison of VMAT lung cancer plans on Halcyon Accelerators using Monte Carlo, Acuros XB, and AAA algorithmsPLOS ONE

Dear Dr. Du,

Thank you for submitting your manuscript to PLOS ONE. After careful consideration, we feel that it has merit but does not fully meet PLOS ONE’s publication criteria as it currently stands. Therefore, we invite you to submit a revised version of the manuscript that addresses the points raised during the review process.

We look forward to receiving your revised manuscript.

Kind regards,

Minsoo Chun, Ph.D.

Academic Editor

PLOS ONE

**Journal Requirements:**

1. When submitting your revision, we need you to address these additional requirements. Please ensure that your manuscript meets PLOS ONE's style requirements, including those for file naming. The PLOS ONE style templates can be found at https://journals.plos.org/plosone/s/file?id=wjVg/PLOSOne_formatting_sample_main_body.pdf and https://journals.plos.org/plosone/s/file?id=ba62/PLOSOne_formatting_sample_title_authors_affiliations.pdf 2. Please note that PLOS ONE has specific guidelines on code sharing for submissions in which author-generated code underpins the findings in the manuscript. In these cases, we expect all author-generated code to be made available without restrictions upon publication of the work. Please review our guidelines at https://journals.plos.org/plosone/s/materials-and-software-sharing#loc-sharing-code and ensure that your code is shared in a way that follows best practice and facilitates reproducibility and reuse. 3. Thank you for stating in your Funding Statement: This study was supported by Zhejiang Provincial Basic Public Welfare Research Project (No. LGF22H160070) and Zhejiang Medical and Health Project (2022KY673).  Please provide an amended statement that declares *all* the funding or sources of support (whether external or internal to your organization) received during this study, as detailed online in our guide for authors at http://journals.plos.org/plosone/s/submit-now.  Please also include the statement “There was no additional external funding received for this study.” in your updated Funding Statement. Please include your amended Funding Statement within your cover letter. We will change the online submission form on your behalf. 4. PLOS requires an ORCID iD for the corresponding author in Editorial Manager on papers submitted after December 6th, 2016. Please ensure that you have an ORCID iD and that it is validated in Editorial Manager. To do this, go to ‘Update my Information’ (in the upper left-hand corner of the main menu), and click on the Fetch/Validate link next to the ORCID field. This will take you to the ORCID site and allow you to create a new iD or authenticate a pre-existing iD in Editorial Manager. 5. Please amend either the abstract on the online submission form (via Edit Submission) or the abstract in the manuscript so that they are identical. 6. We note that there is identifying data in the Supporting Information file “patients_data”. Due to the inclusion of these potentially identifying data, we have removed this file from your file inventory. Prior to sharing human research participant data, authors should consult with an ethics committee to ensure data are shared in accordance with participant consent and all applicable local laws. Data sharing should never compromise participant privacy. It is therefore not appropriate to publicly share personally identifiable data on human research participants. The following are examples of data that should not be shared: -Name, initials, physical address-Ages more specific than whole numbers-Internet protocol (IP) address-Specific dates (birth dates, death dates, examination dates, etc.)-Contact information such as phone number or email address-Location data-ID numbers that seem specific (long numbers, include initials, titled “Hospital ID”) rather than random (small numbers in numerical order) Data that are not directly identifying may also be inappropriate to share, as in combination they can become identifying. For example, data collected from a small group of participants, vulnerable populations, or private groups should not be shared if they involve indirect identifiers (such as sex, ethnicity, location, etc.) that may risk the identification of study participants. Additional guidance on preparing raw data for publication can be found in our Data Policy (https://journals.plos.org/plosone/s/data-availability#loc-human-research-participant-data-and-other-sensitive-data) and in the following article: http://www.bmj.com/content/340/bmj.c181.long. Please remove or anonymize all personal information (<specific identifying information in file to be removed>), ensure that the data shared are in accordance with participant consent, and re-upload a fully anonymized data set. Please note that spreadsheet columns with personal information must be removed and not hidden as all hidden columns will appear in the published file.

Reviewers' comments:

Reviewer's Responses to Questions

**Comments to the Author**

1. Is the manuscript technically sound, and do the data support the conclusions?

Reviewer #1: Yes

Reviewer #2: Partly

2. Has the statistical analysis been performed appropriately and rigorously? 

Reviewer #1: Yes

Reviewer #2: Yes

3. Have the authors made all data underlying the findings in their manuscript fully available?

Reviewer #1: Yes

Reviewer #2: Yes

4. Is the manuscript presented in an intelligible fashion and written in standard English?

Reviewer #1: Yes

Reviewer #2: Yes

5. Review Comments to the Author

**Reviewer #1:** Title: Dosimetric comparison of VMAT lung cancer plans on Halcyon Accelerators using Monte Carlo, Acuros XB, and AAA algorithms

General comments:

This manuscript aims to compare dose distributions calculated by using the RayStation based on Monte Carlo (MC), Anisotropic Analytical Algorithm (AAA), and Acuros XB (AXB) to ensure dose calculation accuracy across different platforms. They also wanted to report dosimetric impact of choosing calculation algorithm on patient dose expectation. The TPS should be fast as possible while it ensures acceptable accuracy compatible to the MC, which has been regarded as a golden standard but requires large amount of time to calculate patient dose distribution. It is essential to validate the accuracy of TPS by comparing the MC simulation. Since many algorithms have been used in the medical institutions, in addition, guarantee of accurate dose calculation across different algorithms is necessary. However, there have been studies on comparing the MC, AAA, and AXB for several years even the beam or tumor site is different (Zaman, et al., 2019, Sarin et al, 2022, Tsuruta, et al., 2014, Han et al., 2011, Seniwal et al., 2020). Authors should suggest their unique findings different from previous studies, but I’m not sure with the current manuscript. I hope they provide them in the next revision of the manuscript or it may be better to submit the manuscript as a technical note.

Specific comments:

- Avoid using abbreviations in the Title.

- Use of RayStation must accompany with using Eclipse to apply the plans to Halcyon machine. Then, is there any advantages of using both the TPSs rather than solely using the Eclipse if the recalculation with the Eclipse must be carried out?

- The abbreviation “haMC”, “haAAA”, and “haAXB” were explained in the Results. As they appears from Materials and Methods, they should be explained its first appearance.

- Even though the RayStation provides MC based calculation, I’m not sure it is compatible with the full-MC simulation as it seems to have adjusted algorithm to make the calculation faster. There may be assumptions to increase calculation speed.

- According to my knowledge, the full-MC has been considered as a gold standard. I wonder that the RayStation can be considered same as the full-MC simulations using the codes such as EGS, Geant4, TOPAS, etc. I think it would be better to include Full-MC simulation result.

- It would be better to provide the required calculation time for obtaining dose distribution with the RayStation since it was considered as the MC calculation and authors explained statistical uncertainty and size of the calculation grid.

- The authors defined the ROIs in the water phantom with different radii, 3 and 5 cm. Is the reason for this to consider both high and low dose region?

- The authors should also provide statistical constraints in the MC calculation in the water phantom study as did they in patient case study.

- The authors categorized their cohort according to tumor stage. Is there any reason for this? This manuscript is not related to treatment outcomes or other biological effects.

- Authors stated they used three algorithms: haMC, haAXB, and haAAA. However, I guess these are labels of data, not the name of the algorithm. I think they should be changed to MC, AXB, and AAA.

- According to the authors, did the AXB also produce dose deviations in boundary regions since the gamma pass rate of AXB&AAA was higher than that of MC&AXB? The gamma pass rate was remarkably increased and this needs to be clarified.

- It would be better to provide gamma analysis for the patient case study similar to the water phantom study.

- What about providing graphs for stating distribution of the parameters such as CI, HI, GI, NTID?

- Please use the same color for the same organ in Figure 12.

**Reviewer #2: **Dosimetric comparison of VMAT lung cancer plans on Halcyon Accelerators using Monte Carlo, Acuros XB, and AAA algorithms

Manuscript Description

The submitted manuscript presents a valuable comparison of dose calculation algorithms—RayStation’s Monte Carlo (MC) and Eclipse’s Acuros XB (AXB) and AAA—in the context of Halcyon VMAT plans for lung cancer treatment. By recalculating treatment plans originally computed with Raystation’s Monte Carlo algorithm using both AcurosXB and AAA, the authors directly compare these algorithms. Their work demonstrates good agreement between MC and AXB across multiple dosimetric metrics in clinical patient cases while AAA shows slight but clinically acceptable deviations. The study’s strengths include the use of a real patient cohort in addition to a simple water phantom and rigorous statistical analysis. Overall, this manuscript makes a noteworthy contribution by demonstrating the reliability of algorithmic transitions for lung VMAT planning on the Halcyon platform.

However, the manuscript requires some major revisions before it can be considered for publication. I offer the following suggestions for revision:

General Comments

1. The introduction should provide a more compelling rationale for the research work. Especially, making it clear that this study is filling in the gap of the work Saini et al. (2021) would be important to show the originality of the study. Also, much of the discussion section about algorithm comparisons needs to be in the introduction, giving previous results as background knowledge.

2. The reasoning behind recalculating without dose renormalization for the prescription dose to cover 95% of the PTV does not seem persuasive enough. The method that “users at other institutions” use seem definitely more acceptable. Further argument backed up with references that support the claim that the CTV still receives the prescribed dose would be helpful for justifying the method used in the study.

Specific Comments

• Halcyon prefixes: Choose either “haMC/haAXB/haAAA” or “MC/AXB/AAA,” and define the “ha” prefix if retained.

• CI vs PCI: Replace “CI” with “PCI” to accurately denote the Paddick Conformity Index to avoid confusion

• Decimal formatting: In tables, ensure consistent decimal formatting (e.g., one decimal place).

• Relative difference calculation: Express relative differences as (Comparing value−Reference)/Reference×100%.

• Consistent Reference Choose an algorithm to serve as the reference (e.g., haMC) and apply it uniformly across all comparisons.

• (p. 1, l. 4): Cite a source to support “VMAT has emerged as a key option…”

• (p. 2, l. 34): Add a reference when comparing AAA to Pencil Beam Convolution.

• (p. 4, l. 127): Include manufacturer name and location for the CT simulator.

• Figure 2: Revisit PTV expansion for Patients 22, 55, and 61—small CTVs appear to have disproportionately large PTVs.

• (p. 5, l. 203): The mentioned NTID isn’t shown in the OAR result

• Table 3 typo: Correct “ClClinical” to “Clinical"

6. PLOS authors have the option to publish the peer review history of their article (what does this mean?). If published, this will include your full peer review and any attached files.

Reviewer #1: No

Reviewer #2: No

---

## [Author Response · Author response to Decision Letter 1]

1 Aug 2025

Included in attached file "Response_to_reviewers.pdf"

---

## [Decision Letter · Decision Letter 1]

25 Aug 2025

PONE-D-25-21057R1Dosimetric Comparison of Monte Carlo, Acuros XB, and Anisotropic Analytical Algorithm for Lung Cancer Plans on Halcyon AcceleratorsPLOS ONE

Dear Dr. Du,

Thank you for submitting your manuscript to PLOS ONE. After careful consideration, we feel that it has merit but does not fully meet PLOS ONE’s publication criteria as it currently stands. Therefore, we invite you to submit a revised version of the manuscript that addresses the points raised during the review process.

We look forward to receiving your revised manuscript.

Kind regards,

Minsoo Chun, Ph.D.

Academic Editor

PLOS ONE

Journal Requirements:

Additional Editor Comments:

The manuscript is generally well-prepared with sufficient data and analysis. However, to further improve the clarity and completeness of the work, the following points should be addressed:

1. Ensure consistent terminology throughout the manuscript (MC, AXB, AAA).

2. More clearly highlight the novelty and contribution of this study compared with prior works.

3. Provide stronger clinical interpretation of the numerical differences, explaining their implications in treatment decision-making.

4. Maintain balanced discussion of TPS advantages, acknowledging that integration benefits may vary by institutional setting.

5. Strengthen references, particularly for definitions of indices (HI, GI, NTID) and for statements regarding algorithm characteristics (e.g., the similarity between AXB and AAA).

5. Provide a clearer discussion of specific results where RayStation MC showed closer agreement with AAA than AXB (e.g., NTID or OAR doses), including possible physical or algorithmic reasons for these findings.

7. Revise the conclusion to emphasize not only the summary of results but also the practical clinical applicability of your findings.

Addressing these points will help enhance the clarity, balance, and overall impact of the manuscript.

Reviewers' comments:

Reviewer's Responses to Questions

**Comments to the Author**

1. If the authors have adequately addressed your comments raised in a previous round of review and you feel that this manuscript is now acceptable for publication, you may indicate that here to bypass the “Comments to the Author” section, enter your conflict of interest statement in the “Confidential to Editor” section, and submit your "Accept" recommendation.

Reviewer #1: (No Response)

Reviewer #2: All comments have been addressed

2. Is the manuscript technically sound, and do the data support the conclusions?

Reviewer #1: Yes

Reviewer #2: Yes

3. Has the statistical analysis been performed appropriately and rigorously? 

Reviewer #1: Yes

Reviewer #2: Yes

4. Have the authors made all data underlying the findings in their manuscript fully available?

Reviewer #1: Yes

Reviewer #2: Yes

5. Is the manuscript presented in an intelligible fashion and written in standard English?

Reviewer #1: Yes

Reviewer #2: Yes

6. Review Comments to the Author

Reviewer #1: Title: Dosimetric Comparison of Monte Carlo, Acuros XB, and Anisotropic Analytical Algorithm for Lung Cancer Plans on Halcyon Accelerators

General comments:

The authors have well revised their manuscript and I appreciate their efforts in addressing most of the comments in the first revision. All the inquiries were clarified clearly. The motivation, novelty and strengths of their works were enhanced and clear. However, one point still needs clarification. I still think that the RayStation calculates based on the MC method but it doesn’t represent the MC typically refers to in this research field. It would be more precise to describe the calculation as “RayStation MC” or an equivalent term rather than simply “MC”.

Other comments are listed below as the specific comments.

Specific comments:

- Please specify which accelerator model parameters were adjusted during the TPS commissioning.

- Authors previously responded to the comment on the rationale for using different ROI sizes in water phantom test but their clarification is not sufficient. They should provide the reasons why 3-cm-radious ROI is more appropriate to assess the high dose uniform region rather than 2 or 4 cm. Similarly, they should also clarify why 5-cm-radious ROI is suitable for representing the dose gradient or lower-dose regions than the ROIs with any other sizes. For example, in my assumption, authors could use the ROIs with the most similar sizes for both high or low dose region.

- It would be better to cite more relevant references for definitions or equations of HI and GI.

- Please add reference for convincing that the AXB and AAA share similar dose calculation approximations, because they are implemented in the same TPS.

- According to understanding, the RayStation MC shows better agreement with the AXB than with the AAA. However, some results (e. g. NTID, OAR doses) showed better agreement between the RayStation and AAA than between the RayStation and AXB. The authors should clarify this.

- Authors clarified that one of their strengths over previous researches were that the validation is specified for Halcyon accelerator. I assume there are not substantial differences in beams of the same energy, such as 6 MV FFF. The differences between the accelerator types, C-type or O-type, are not expected to affect the TPS dose calculation accuracy. Could the authors clarify whether significant differences exist?

- Authors suggested one of the advantages of RayStation over Eclipse is integrated workflow. But the Eclipse seems to be more integrated in the case of medical facilities equipped exclusively with Varian accelerators.

Reviewer #2: The authors' response to the reviewers' comments seems to have been well reflected in this revised paper.

7. PLOS authors have the option to publish the peer review history of their article (what does this mean?). If published, this will include your full peer review and any attached files.

Reviewer #1: No

Reviewer #2: No

---

## [Author Response · Author response to Decision Letter 2]

8 Sep 2025

Included in the attached Response_to_reviewers.pdf

---

## [Decision Letter · Decision Letter 2]

26 Sep 2025

Dosimetric Comparison of Monte Carlo, Acuros XB, and Anisotropic Analytical Algorithm for Lung Cancer Plans on Halcyon Accelerators

PONE-D-25-21057R2

Dear Dr. Du,

We’re pleased to inform you that your manuscript has been judged scientifically suitable for publication and will be formally accepted for publication once it meets all outstanding technical requirements.

Kind regards,

Minsoo Chun, Ph.D.

Academic Editor

PLOS ONE

Additional Editor Comments (optional):

I am pleased to accept this manuscript. Ensure that the reviewer’s minor remarks are addressed in the final submission.

Reviewers' comments:

Reviewer's Responses to Questions

**Comments to the Author**

1. If the authors have adequately addressed your comments raised in a previous round of review and you feel that this manuscript is now acceptable for publication, you may indicate that here to bypass the “Comments to the Author” section, enter your conflict of interest statement in the “Confidential to Editor” section, and submit your "Accept" recommendation.

Reviewer #1: All comments have been addressed

Reviewer #2: All comments have been addressed

2. Is the manuscript technically sound, and do the data support the conclusions?

Reviewer #1: Yes

Reviewer #2: Yes

3. Has the statistical analysis been performed appropriately and rigorously? 

Reviewer #1: Yes

Reviewer #2: Yes

4. Have the authors made all data underlying the findings in their manuscript fully available?

Reviewer #1: Yes

Reviewer #2: Yes

5. Is the manuscript presented in an intelligible fashion and written in standard English?

Reviewer #1: Yes

Reviewer #2: Yes

6. Review Comments to the Author

Reviewer #1: I really appreciate your revision with consideration of my suggestions. Now, the manuscript seems to be ready to go on. I don't have any further inquiries or comments on this manuscript.

Reviewer #2: Since the authors have addressed Reviewer 1’s comment by distinguishing between 'Monte Carlo (MC)' and 'RayStation Monte Carlo (RMC),' I suggest also replacing 'haMC' with 'haRMC' to maintain consistency and avoid confusion.

7. PLOS authors have the option to publish the peer review history of their article (what does this mean?). If published, this will include your full peer review and any attached files.

Reviewer #1: No

Reviewer #2: No

---

## [Editor Report · Acceptance letter]

PONE-D-25-21057R2

PLOS ONE

Dear Dr. Du,

I'm pleased to inform you that your manuscript has been deemed suitable for publication in PLOS ONE. Congratulations! Your manuscript is now being handed over to our production team.

Kind regards,

on behalf of

Dr. Minsoo Chun

Academic Editor

PLOS ONE